EMBO
Molecular Medicine

# mTORC2 sustains thermogenesis via Akt-induced glucose uptake and glycolysis in brown adipose tissue

Verena Albert[1], Kristoffer Svensson[1], Mitsugu Shimobayashi[1], Marco Colombi[1], Sergio Muñoz[2,3], Veronica Jimenez[2,3], Christoph Handschin[1], Fatima Bosch[2,3] & Michael N Hall[1,*]

## Abstract

Activation of non-shivering thermogenesis (NST) in brown adipose tissue (BAT) has been proposed as an anti-obesity treatment. Moreover, cold-induced glucose uptake could normalize blood glucose levels in insulin-resistant patients. It is therefore important to identify novel regulators of NST and cold-induced glucose uptake. Mammalian target of rapamycin complex 2 (mTORC2) mediates insulin-stimulated glucose uptake in metabolic tissues, but its role in NST is unknown. We show that mTORC2 is activated in brown adipocytes upon β-adrenergic stimulation. Furthermore, mice lacking mTORC2 specifically in adipose tissue (AdRiKO mice) are hypothermic, display increased sensitivity to cold, and show impaired cold-induced glucose uptake and glycolysis. Restoration of glucose uptake in BAT by overexpression of hexokinase II or activated Akt2 was sufficient to increase body temperature and improve cold tolerance in AdRiKO mice. Thus, mTORC2 in BAT mediates temperature homeostasis via regulation of cold-induced glucose uptake. Our findings demonstrate the importance of glucose metabolism in temperature regulation.

**Keywords** brown adipose tissue; glucose uptake; mTORC2; thermogenesis
**Subject Category** Metabolism

## Introduction

Non-shivering thermogenesis (NST) in brown adipose tissue (BAT) allows mammals to maintain stable body temperature in a cold environment. Upon cold exposure, norepinephrine (NE) is released from sympathetic nerves and binds to adrenergic receptors on brown adipocytes to induce NST. Adrenergic receptor stimulation induces cAMP production and subsequent induction of lipolysis, β-oxidation, and mitochondrial uncoupling (Cannon & Nedergaard, 2004). Mitochondrial uncoupling occurs through activation of uncoupling protein 1 (UCP1). UCP1 is a mitochondrial transmembrane protein specifically expressed in brown adipocytes and brown-like, beige adipocytes. Once activated, UCP1 dissipates the proton gradient across the inner mitochondrial membrane generated by the electron transport chain. This uncouples proton flux into the mitochondria from ATP production, resulting in heat generation (Klaus et al, 1991; Busiello et al, 2015). To compensate for the loss of mitochondrial ATP production due to uncoupling, adrenergic stimulation enhances glucose uptake and glycolysis in BAT (Greco-Perotto et al, 1987; Vallerand et al, 1990; Hao et al, 2015). Due to the ability of BAT to burn energy efficiently and to reduce blood glucose levels, activation of NST has been proposed as an alternative strategy for weight loss in obese patients (Cypess et al, 2009; van Marken Lichtenbelt et al, 2009; Virtanen et al, 2009; Clapham & Arch, 2011) and for normalization of blood glucose levels in insulin-resistant diabetic patients. Thus, identifying novel regulators of NST could provide new drug targets for anti-obesity and diabetes treatments.

The mammalian target of rapamycin (mTOR) signaling network is a central regulator of cell growth and metabolism (Laplante & Sabatini, 2012; Dibble & Manning, 2013; Albert & Hall, 2014; Shimobayashi & Hall, 2014). mTOR is a highly conserved protein kinase found in two structurally and functionally distinct complexes named mTOR complex 1 (mTORC1) and mTORC2. mTORC1 is sensitive to the macrolide rapamycin and contains mTOR, mammalian lethal with sec-13 protein (mLST8), and regulatory associated protein of mTOR (raptor). mTORC2 is rapamycin insensitive and contains mTOR, mLST8, mammalian stress-activated map kinase-interacting protein 1 (mSIN1), and rapamycin-insensitive companion of mTOR (rictor). mTORC2 is activated by growth factors, such as insulin and insulin-like growth factor 1 (IGF-1), via phosphatidylinositol 3-kinase (PI3K)-dependent ribosome association (Zinzalla et al, 2011). mTORC2 downstream targets are members of the AGC kinase family, such as Akt, serum/glucocorticoid-regulated kinase

1  Biozentrum, University of Basel, Basel, Switzerland
2  Center of Animal Biotechnology and Gene Therapy and Department of Biochemistry and Molecular Biology, School of Veterinary Medicine, Universitat Autònoma de Barcelona, Bellaterra, Spain
3  Centro de Investigación Biomédica en Red de Diabetes y Enfermedades Metabólicas Asociadas (CIBERDEM), Madrid, Spain
*Corresponding author. Tel: +41 61 267 21 50; E-mail: m.hall@unibas.ch

    

(SGK), and protein kinase C (PKC) (Sarbassov *et al*, 2005; Jacinto *et al*, 2006; Garcia-Martinez & Alessi, 2008; Ikenoue *et al*, 2008; Cybulski & Hall, 2009), through which mTORC2 promotes lipogenesis, glucose uptake, glycolysis, and cell survival (Manning & Cantley, 2007; Kumar *et al*, 2008; Hagiwara *et al*, 2012; Yuan *et al*, 2012). Due to its role in mediating lipid and glucose homeostasis, dysfunction of mTORC2 signaling has been implicated in the development of insulin resistance and diabetes. Moreover, a recent study by Olsen *et al* (2014) demonstrated that mTORC2 in brown adipocytes *in vitro* mediates β-adrenergic stimulation-induced glucose uptake. However, a role for mTORC2 in thermogenesis, and in particular NST, has so far not been investigated.

Here, we show that β-adrenergic stimulation and cold exposure activate mTORC2 signaling in brown adipocytes *in vitro* and *in vivo*. We find that mTORC2 in BAT stimulates cold-induced glucose uptake and glycolysis. Consequently, mice with adipose tissue-specific inactivation of mTORC2 (AdRiKO mice) are hypothermic and unable to maintain stable body temperature upon cold exposure. Restoration of either Akt signaling or glucose metabolism in BAT of AdRiKO mice restored body temperature and improved cold tolerance. Thus, mTORC2 in BAT is essential for maintenance of energy homeostasis and body temperature upon cold exposure.

## Results

### Norepinephrine activates mTORC2 *in vitro* via cAMP, PI3K, and Epac1

To investigate the role of mTORC2 signaling in NST, we first examined whether mTORC2 is activated by signals that induce thermogenesis. In particular, differentiated brown adipocytes (dBACs) were treated with NE to induce β-adrenergic signaling. As expected, NE treatment resulted in stimulation of PKA signaling as suggested by increased Creb-S133, HSL-S563, and perilipin phosphorylation (Fig 1A). Importantly, NE also stimulated phosphorylation of the mTORC2 target Akt at S473, a major readout of mTORC2 activity (Hresko & Mueckler, 2005; Sarbassov *et al*, 2005; Cybulski & Hall, 2009) (Fig 1A). Moreover, NE stimulation also induced phosphorylation of mTOR at S2481, another indicator of mTORC2 activation (Copp *et al*, 2009). These observations suggest that β-adrenergic stimulation induces mTORC2 signaling, in addition to PKA, in dBACs.

Next, we investigated the pathway via which NE stimulates mTORC2. Insulin activates mTORC2 in a PI3K-dependent but mTORC1-independent manner. We examined whether NE activates mTORC2 in a similar manner. We stimulated dBACs with NE in the presence of the pan-mTOR (mTORC1 and mTORC2) inhibitor Torin, the mTORC1-specific inhibitor rapamycin, or the PI3K inhibitor wortmannin. Similar to insulin-induced mTORC2 stimulation, NE-induced activation of mTORC2 was independent of mTORC1, since pretreatment of dBACs with rapamycin did not prevent induction of Akt-S473 phosphorylation upon NE stimulation (Fig 1B). In contrast, inhibition of mTOR with Torin or of PI3K with wortmannin prevented Akt-S473 phosphorylation (Fig 1B). Hence, NE-induced activation of mTORC2 in dBACs is dependent on PI3K and independent on mTORC1.

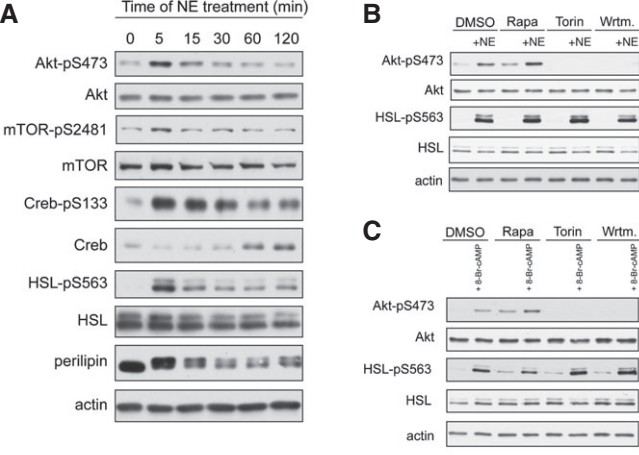

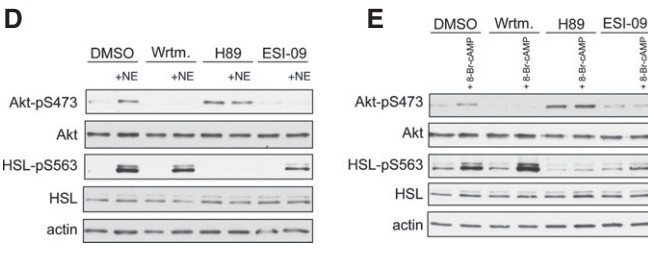

**Figure 1. NE activates mTORC2 *in vitro* via cAMP, PI3K, and Epac1.**

A Immunoblot analysis of BAT cells stimulated with norepinephrine (NE) for the indicated proteins.

B Immunoblot analysis of BAT cells stimulated with NE for 5 min in the presence of rapamycin (Rapa), Torin, or wortmannin (Wrtm) for the indicated proteins.

C Immunoblot analysis of BAT cells stimulated with 8-Br-cAMP for 5 min in the presence of Rapa, Torin, or Wrtm for the indicated proteins.

D Immunoblot analysis of BAT cells stimulated with NE for 5 min in the presence of Wrtm, H89, or ESI-09 for the indicated proteins.

E Immunoblot analysis of BAT cells stimulated with 8-Br-cAMP for 5 min in the presence of Wrtm, H89, or ESI-09 for the indicated proteins.

Data information: All experiments were performed in triplicates, and a representative replicate is presented.

NE stimulation leads to an increase in intracellular cAMP, which is crucial for NE-induced activation of PKA signaling (Cannon & Nedergaard, 2004). To test whether cAMP is required for NE-induced activation of mTORC2, we treated dBACs with the cell-permeable cAMP analogue 8-Br-cAMP. Similar to NE stimulation, 8-Br-cAMP treatment induced Akt-S473 phosphorylation in dBACs (Fig 1C). 8-Br-cAMP stimulated mTORC2 signaling when mTORC1 was blocked with rapamycin, but was no longer able to induce Akt-S473 phosphorylation when mTOR or PI3K was inhibited with Torin or wortmannin, respectively (Fig 1C). Thus, NE induces mTORC2 signaling via cAMP and PI3K. Importantly, in the presence of mTOR or PI3K inhibition, NE or 8-Br-cAMP still induced HSL-S563 phosphorylation (Fig 1B and C), indicating that inhibition of mTOR or PI3K does not affect PKA signaling.

cAMP has several target proteins, two of which are PKA and Epac1. Epac1 mediates cAMP-induced activation of mTORC2 in prostate cancer cells (Misra & Pizzo, 2012) and thus might be

involved in cAMP-induced stimulation of mTORC2 in BAT. To investigate whether PKA or Epac1 is required for NE- or cAMP-induced activation of mTORC2, we stimulated dBACs with NE or 8-Br-cAMP in the presence of the PKA inhibitor H89 or the Epac inhibitor ESI-09. Treatment of dBACs with H89 efficiently blocked NE- and 8-Br-cAMP-induced HSL-S563 phosphorylation, suggesting that PKA signaling is inhibited. ESI-09 treatment slightly reduced but did not block induction of HSL-S563 phosphorylation by NE or 8-Br-cAMP stimulation (Fig 1D and E). Interestingly, treatment of dBACs with H89 resulted in hyperphosphorylation of Akt-S473, suggesting that inhibition of PKA signaling does not impair activation of mTORC2 (Fig 1D and E). In contrast, treatment with the Epac inhibitor ESI-09 prevented NE- and 8-Br-cAMP-induced phosphorylation of Akt-S473, suggesting that NE activates mTORC2 via cAMP, Epac1, and PI3K (Fig 1D and E).

## Norepinephrine and cold activate mTORC2 *in vivo*

We next assessed whether NE can stimulate mTORC2 signaling in BAT *in vivo*. To this end, we used AdRiKO mice (Cybulski *et al*, 2009) which are defective in mTORC2 signaling in both BAT and white adipose tissue (WAT) (Fig EV1A). AdRiKO mice display increased lean mass and elevated insulin-like growth factor 1 (IGF-1) levels upon high-fat diet (Cybulski *et al*, 2009). To avoid confounding effects due to this growth phenotype, we used young (10–14 weeks) AdRiKO mice fed a standard diet. Under such conditions, AdRiKO mice are not altered in body weight, body composition, and circulating IGF-1 levels (Fig EV1B–D). In line with our *in vitro* results, treatment of control mice with NE induced phosphorylation of the mTORC2 target Akt and of S2481 on mTOR (Fig 2A). Importantly, AdRiKO mice did not display induction of Akt-S473 phosphorylation in BAT upon NE stimulation (Figs 2B and EV1E). Hence, functional mTORC2 is required for Akt phosphorylation in BAT in response to NE. Since NE is released from sympathetic nerves upon cold exposure, we hypothesized that mTORC2 signaling in BAT could also be induced by cold stress. Similar to the results obtained with NE stimulation, cold exposure induced Akt-S473 and mTOR-S2481 phosphorylation in BAT of control mice (Figs 2C and EV1F). Again, this induction was dependent on functional mTORC2 signaling as Akt-pS473, mTOR-pS2481, and phosphorylation of the Akt target FoxO1 were not induced in BAT upon cold exposure of AdRiKO mice (Figs 2C and EV1F). In contrast to BAT, mTORC2 signaling was not induced in inguinal subcutaneous WAT (sWAT) upon cold exposure (Fig EV1G). Taken together, these data demonstrate that mTORC2 signaling is induced by NE and cold in BAT but not in sWAT.

As we observed an induction of mTORC2 signaling in BAT upon NE and cold stimulation, we next investigated whether a defect in mTORC2 signaling affected temperature regulation. AdRiKO mice were hypothermic when housed at 22°C, which is a mild temperature stress for mice (Fig 2D). The hypothermia could not be accounted for by a reduction in locomotor activity (Fig EV1H). In contrast, housing AdRiKO mice at thermoneutrality (30°C) for 2 weeks prevented this hypothermic phenotype (Fig 2E). Next, we performed an acute cold exposure with AdRiKO and control mice. In contrast to control mice, AdRiKO mice were unable to maintain stable body temperature when housed at 4°C (Fig 2F). Interestingly, when food was provided during acute cold

exposure, AdRiKO mice displayed a less severe loss of body temperature, but were still unable to maintain stable body temperature in the cold (Fig EV1I). Thus, inactivation of mTORC2 signaling in adipose tissue leads to decreased body temperature and increased sensitivity to cold stress, in particular under nutrient-limiting conditions.

Cold-induced muscle shivering also contributes to heat generation upon acute cold exposure (Cannon & Nedergaard, 2004). To investigate whether the increased sensitivity to cold stress observed in AdRiKO mice was due to impaired shivering thermogenesis, we measured cold-induced muscle shivering in AdRiKO and control mice after 4-h cold exposure. Interestingly, cold-exposed AdRiKO mice showed significantly increased cold-induced muscle shivering compared to control mice (Fig EV1J). The increased shivering could be a compensatory reaction of the AdRiKO mice to maintain body temperature upon cold stress.

## mTORC2 in adipose tissue is not required for cold-induced lipid droplet mobilization, mitochondrial uncoupling, and β-oxidation

Thermogenesis upon β-adrenergic stimulation requires mobilization of lipid stores, induction of β-oxidation, and stimulation of mitochondrial uncoupling to generate heat. Since AdRiKO mice are hypothermic and exhibit increased sensitivity to cold (see above), we investigated whether AdRiKO mice are defective in lipid mobilization, β-oxidation, or mitochondrial uncoupling in adipose tissue. There was no difference in BAT and sWAT weights between AdRiKO and control mice housed at either 22 or 4°C (Fig EV2A and B). Moreover, there was no discernible difference in the morphology of lipid droplets in sWAT from AdRiKO mice compared to wild-type control mice kept at 22°C. Furthermore, both control and AdRiKO mice were able to mobilize sWAT lipid stores upon cold exposure (4°C) as suggested by a reduction in the size of lipid droplets, that is, the appearance of multilocular adipocytes in sWAT (Fig 3A). In line with this, cold-exposed AdRiKO and control mice both displayed a significant increase in levels of circulating free fatty acids (NEFAs) and glycerol (Fig 3B and C). Even though the increase in circulating NEFAs was slightly less in AdRiKO mice compared to control mice, AdRiKO mice still displayed a twofold increase in circulating NEFAs upon cold exposure (Fig 3B). In BAT, AdRiKO mice housed at 22°C displayed larger lipid droplets compared to control mice (Fig 3D). However, at 4°C lipid droplet size in BAT decreased to the same extent in AdRiKO and control mice (Fig 3D). Despite the difference in lipid droplet size, we did not observe a significant difference in total triglyceride (TG) content in BAT upon cold exposure or between control and AdRiKO mice (Fig 3E). In contrast to this, free fatty acid levels in BAT were strongly enhanced in the AdRiKO mice upon cold exposure (Fig 3F). These findings suggest that the defect in temperature regulation in AdRiKO mice is most likely not due to decreased availability of free fatty acids upon cold exposure.

Since cold-exposed AdRiKO mice display significantly increased levels of free fatty acids in BAT compared to control mice (Fig 3F), we hypothesized that this increase in NEFAs might be due to impaired mitochondrial function, which could lead to accumulation of NEFAs in BAT. To test this possibility, we first measured induction of genes involved in mitochondrial uncoupling in BAT upon cold exposure. Despite the cold-sensitive phenotype of AdRiKO

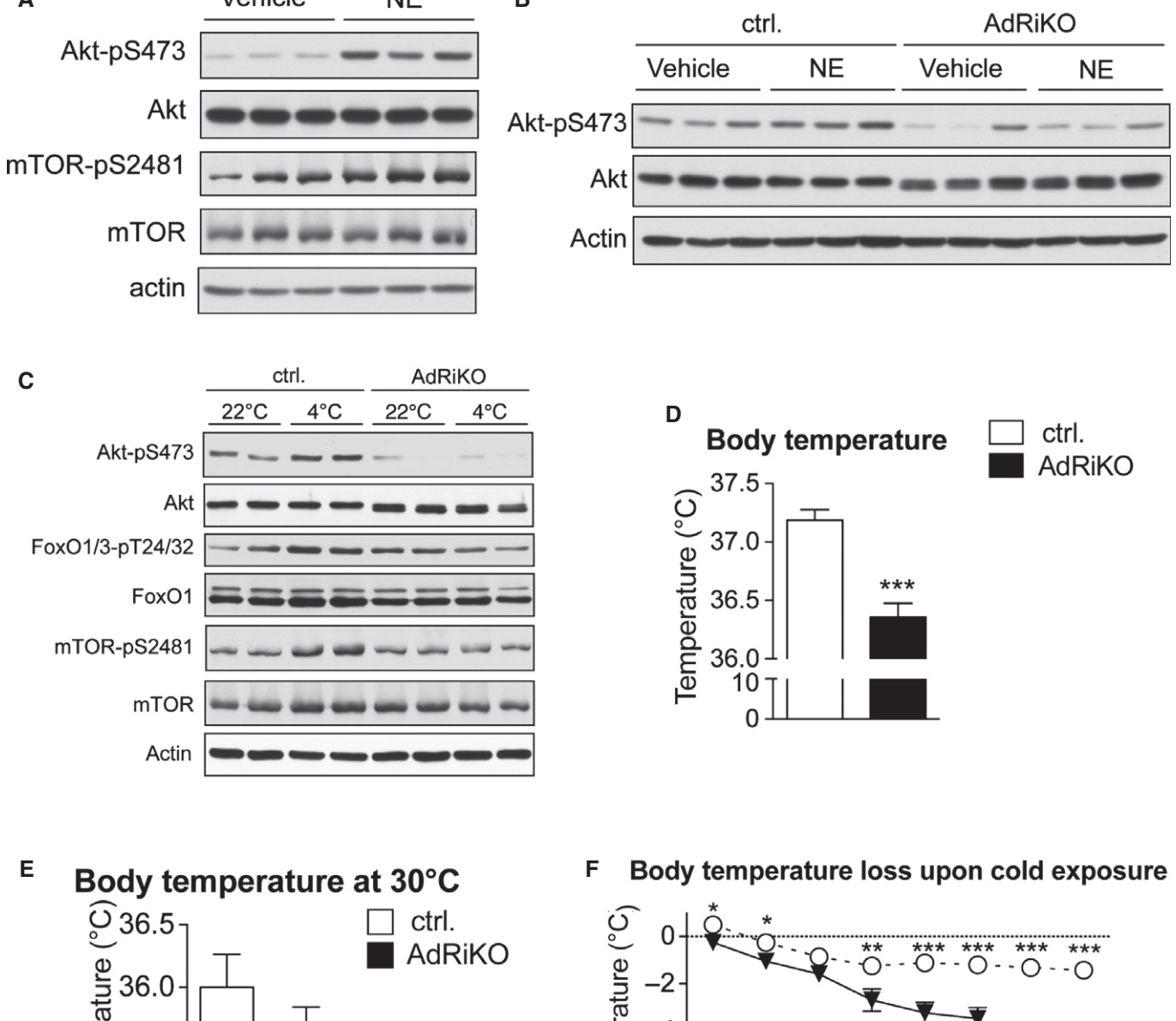

**Figure 2. NE and cold activate mTORC2 *in vivo*.**

A Immunoblot analysis of BAT from control mice treated with either norepinephrine (NE) or vehicle for 30 min for the indicated proteins ($n$ = 3/group).

B Immunoblot analysis of BAT from AdRiKO and control mice treated with either norepinephrine (NE) or vehicle for 30 min for the indicated proteins ($n$ = 3/group).

C Immunoblot analysis of BAT from AdRiKO and control mice housed at either 22 or 4°C for 2 h for the indicated proteins ($n$ = 6/group, each lane represents a mix of 3 mice).

D Body temperature of AdRiKO and control mice housed at 22°C [$n$ = 11 (control), $n$ = 9 (AdRiKO)].

E Body temperature of AdRiKO and control mice housed at 30°C for 2 weeks ($n$ = 8/group).

F Body temperature loss upon cold exposure of AdRiKO and control mice [$n$ = 20 (control), $n$ = 17 (AdRiKO)].

Data information: Data represent mean ± SEM. Statistically significant differences between AdRiKO and control mice were determined with unpaired Student's *t*-test and are indicated with asterisks (*$P$ < 0.05; **$P$ < 0.01; ***$P$ < 0.001). The exact *P*-value for each significant difference can be found in Appendix Table S2.

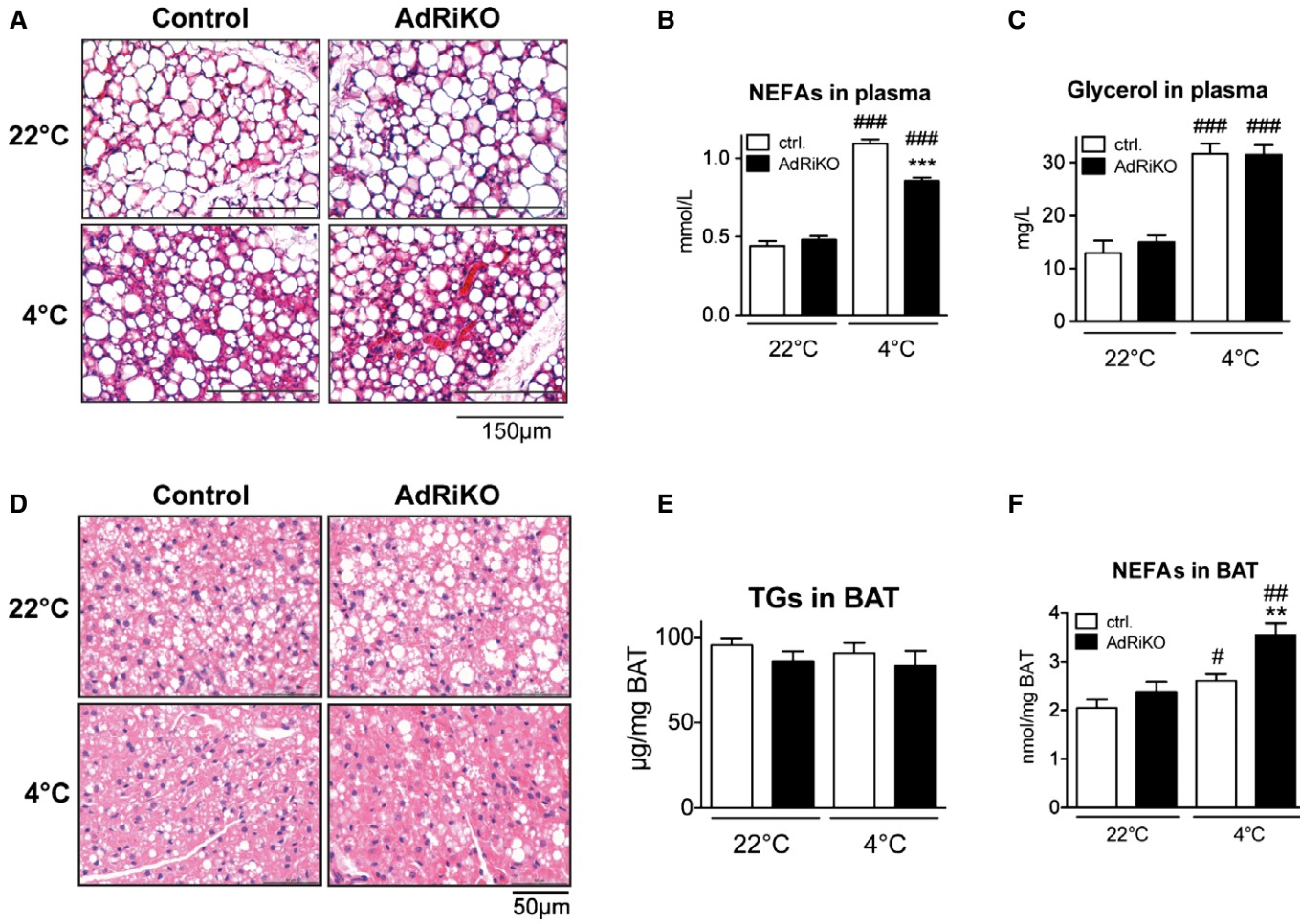

**Figure 3.  mTORC2 in adipose tissue is not required for cold-induced lipid droplet mobilization.**

A    Representative H&E staining of sWAT sections from AdRiKO and control mice (*n* = 5/group).
B    Non-esterified fatty acids (NEFAs) in plasma of AdRiKO and control mice (*n* = 6/group).
C    Glycerol in plasma of AdRiKO and control mice (*n* = 6/group).
D    Representative H&E staining of BAT sections from AdRiKO and control mice (*n* = 5/group).
E    Triglycerides (TGs) in BAT of AdRiKO and control mice housed at 22 or 4°C for 8 h (*n* = 6/group).
F    NEFAs in BAT of AdRiKO and control mice (*n* = 6/group).

Data information: Data represent mean ± SEM. Statistically significant differences between AdRiKO and control mice were determined with unpaired Student's *t*-test and are indicated with asterisks (**$P < 0.01$; ***$P < 0.001$). Statistically significant differences between temperatures were determined with unpaired Student's *t*-test and are indicated with a number sign (#$P < 0.05$; ##$P < 0.01$; ###$P < 0.001$). The exact *P*-value for each significant difference can be found in Appendix Table S2.

mice, mRNA levels of *UCP1*, *Dio2*, and *PGC-1α* were induced to a similar extent in BAT of AdRiKO and control mice (Fig 4A), and UCP1 protein levels in BAT were also similar (Fig 4B). Second, AdRiKO mice did not exhibit any defect in expression of genes involved in β-oxidation (Fig 4C). Thus, AdRiKO mice appear normal for induction of the thermogenic transcriptional program and expression of β-oxidation genes. Third, we measured expression of proteins of the electron transport chain in BAT. AdRiKO mice displayed a slight decrease (22°C) or no change (4°C) in expression of electron transport chain proteins compared to control mice (Fig 4D). Fourth, mitochondrial DNA (mtDNA) copy number was unchanged in BAT of AdRiKO mice (Fig 4E), suggesting that BAT of AdRiKO and control mice contain a similar amount of mitochondria. Fifth, EM micrographs of BAT revealed no difference between

AdRiKO and control mitochondria with regard to size, shape, and cristae structure (Fig 4F). Finally, cold-exposed AdRiKO mice exhibited normal induction of oxygen consumption in BAT (Fig 4G) and at the whole-body level (Fig 4H). Thus, BAT in AdRiKO mice has normal mitochondrial function and oxidative metabolism can be efficiently induced upon cold stress. This suggests that the observed cold sensitivity of AdRiKO mice does not stem from a mitochondrial defect.

Despite a similar maximal induction of whole-body respiration (Fig 4H), AdRiKO mice were unable to maintain an enhanced metabolic rate throughout the duration of the cold exposure time course (Fig 4I). This inability to maintain an enhanced metabolic rate may account for the inability of AdRiKO mice to sustain an NST response.

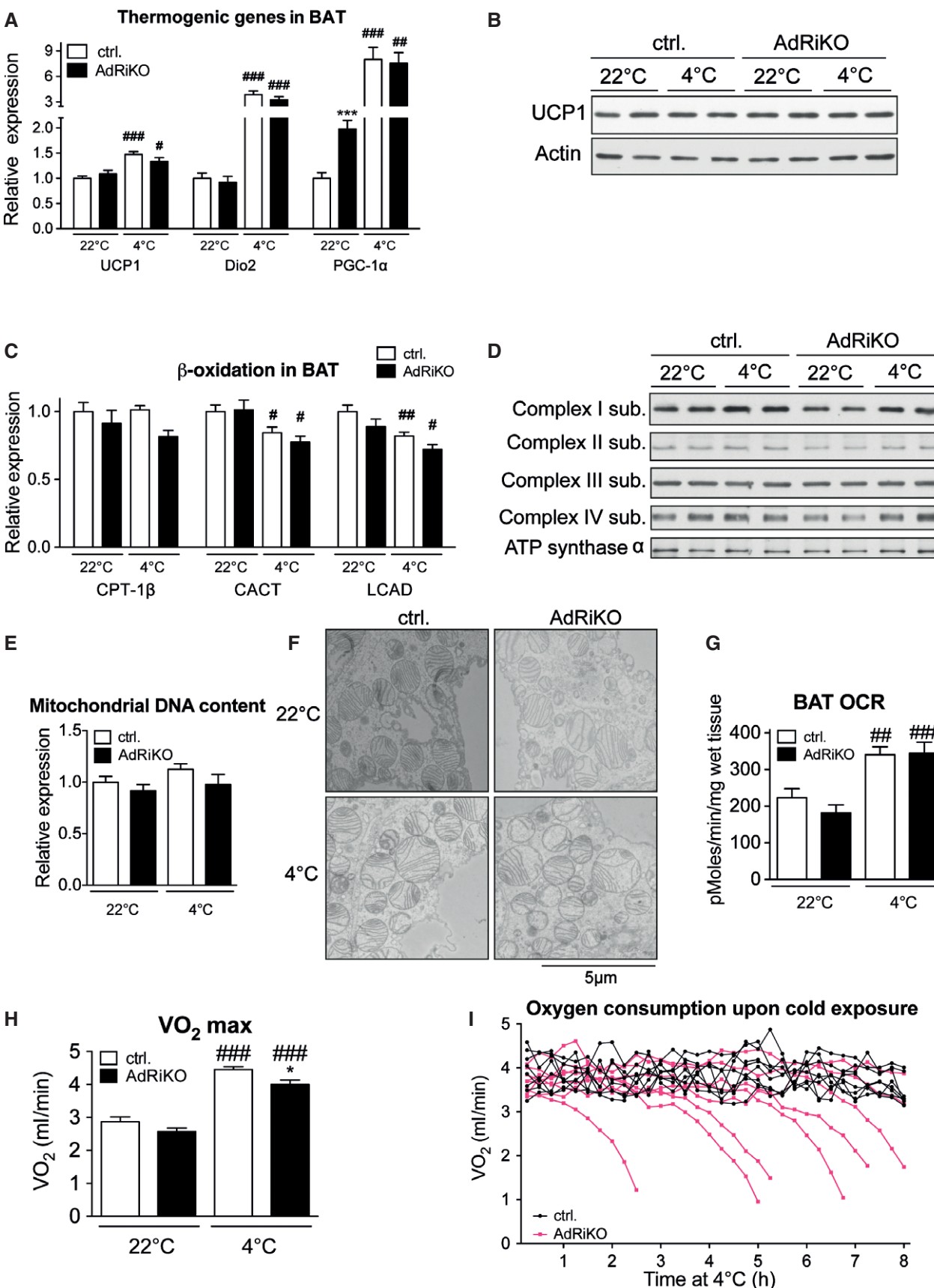

**Figure 4.**

**Figure 4.  mTORC2 in adipose tissue is not required for cold-induced mitochondrial uncoupling and β-oxidation.**

A    mRNA levels of the indicated genes in BAT of AdRiKO and control mice housed at 22 or at 4°C for 8 h ($n$ = 6/group).
B    Immunoblot analysis of BAT from AdRiKO and control mice housed at 22 or 4°C for 8 h for the indicated proteins ($n$ = 6/group, each lane represents a mix of 3 mice).
C    mRNA levels of the indicated genes in BAT of AdRiKO and control mice housed at 22 or at 4°C for 8 h ($n$ = 6).
D    Immunoblot analysis of BAT from AdRiKO and control mice housed at 22 or 4°C for 8 h for the indicated proteins ($n$ = 6/group, each lane represents a mix of 3 mice).
E    Mitochondrial DNA content of BAT from AdRiKO and control mice housed at 22 or at 4°C for 8 h ($n$ = 6/group).
F    Representative electron micrographs of BAT from AdRiKO and control mice housed at 22 or at 4°C for 4 h ($n$ = 3/group).
G    Oxygen consumption rate (OCR) of BAT explants from AdRiKO and control mice housed at 22 or at 4°C for 4 h ($n$ = 7/group).
H    Maximal respiration (VO$_2$ max) of AdRiKO and control mice housed at 22 or at 4°C for 8 h [$n$ = 9 (control 22°C), $n$ = 7 (AdRiKO 22°C), $n$ = 8 (control 4°C), $n$ = 8 (AdRiKO 4°C)].
I    Respiration (VO$_2$) of AdRiKO and control mice upon cold exposure ($n$ = 8/group).

Data information: Data represent mean ± SEM. Statistically significant differences between AdRiKO and control mice were determined with unpaired Student's $t$-test and are indicated with asterisks (*$P$ < 0.05; ***$P$ < 0.001). Statistically significant differences between temperatures were determined with unpaired Student's $t$-test and are indicated with a number sign ($^{\#}P$ < 0.05; $^{\#\#}P$ < 0.01; $^{\#\#\#}P$ < 0.001). The exact $P$-value for each significant difference can be found in Appendix Table S2.

## mTORC2 in adipose tissue is required for cold-induced glucose uptake and glycolysis

Glucose uptake and glycolysis are strongly enhanced in BAT upon cold exposure, to compensate for the loss of mitochondrial ATP production due to heat-generating mitochondrial uncoupling (Greco-Perotto *et al*, 1987; Vallerand *et al*, 1990; Hao *et al*, 2015). Moreover, glucose is also needed for anaplerotic reactions to maintain fatty acid oxidation, and to generate glycerol 3-phosphate for lipid synthesis. mTORC2 is an important regulator of insulin-induced glucose uptake and glycolysis in WAT, muscle, and liver (Kumar *et al*, 2008, 2010; Hagiwara *et al*, 2012). Thus, reduced glucose uptake and glycolysis might explain the failure of AdRiKO mice to maintain an enhanced metabolic rate upon cold exposure (Fig 4I). To test this notion, we examined cold-induced glucose uptake in BAT. More specifically, we measured 2-deoxyglucose-6-phosphate (2DG6P) accumulation in BAT 45 min after injecting mice with 2-deoxyglucose (2DG). BAT in AdRiKO mice displayed significantly impaired glucose uptake upon cold exposure (Fig 5A). Similarly, cold-exposed AdRiKO mice failed to increase glycolysis in BAT, as indicated by a reduced extracellular acidification rate (ECAR) in BAT explants (Fig 5B). Thus, mTORC2 signaling is required to induce glucose uptake and glycolysis in BAT upon cold exposure.

To further investigate glucose homeostasis in AdRiKO and control mice, we measured blood glucose and plasma insulin levels. Both AdRiKO and control mice displayed a significant decrease in blood glucose upon cold exposure (Fig EV3A). Interestingly, cold-exposed AdRiKO mice displayed slightly lower blood glucose compared to cold-exposed control mice (Fig EV3A), despite impaired glucose uptake into BAT (Fig 5A). This decrease might be due to increased shivering and muscle activity in AdRiKO mice upon cold exposure (Fig EV1J). Next, we assessed levels of circulating insulin. As previously shown (Cybulski *et al*, 2009), AdRiKO mice displayed hyperinsulinemia when kept at 22°C (Fig EV3B). Upon cold exposure, we observed a significant drop in plasma insulin levels which reached a comparable value in both AdRiKO and control mice (Fig EV3B). Thus, changes in circulating insulin are unlikely to explain the defect in cold-induced glucose uptake in BAT in AdRiKO mice.

To determine whether the absence of mTORC2 in BAT leads to energetic stress upon cold exposure, which could account for

an inability to sustain NST, we examined AMP-activated protein kinase (AMPK) signaling. AMPK is activated in response to low energy levels (Hardie & Hawley, 2001). AdRiKO mice displayed enhanced phosphorylation of the AMPK targets ACC and raptor (Figs 5C and EV3C), indicating energy stress in BAT of these mice. Interestingly, we also observed a higher molecular weight isoform of AMPK specifically in cold-exposed AdRiKO mice (Fig 5C).

Next, we investigated how mTORC2 signaling affects glucose uptake and glycolysis in BAT. mTORC2 signaling has been shown to mediate insulin-stimulated translocation of GLUT4 to the plasma membrane (Kumar *et al*, 2010). Moreover, GLUT1 is involved in glucose uptake into brown adipocytes upon adrenergic stimulation (Dallner *et al*, 2006). To test whether mTORC2 signaling in BAT affects plasma membrane localization of GLUT1 or GLUT4, we isolated plasma membrane from AdRiKO and control BAT. The amount of GLUT1 and GLUT4 in the two plasma membrane fractions was similar (Fig 5D). This suggests that mTORC2 in BAT does not mediate cold-induced glucose uptake and glycolysis by affecting localization of glucose transporters. Glucose uptake is also affected by hexokinases, which phosphorylate glucose to catalyze the first and rate-limiting step of glycolysis. Of the four different hexokinase isoforms, hexokinase I (HKI) and hexokinase II (HKII) are the two dominant isoforms in BAT and are found both in the cytosol and at mitochondria (Shinohara *et al*, 1998; Wilson, 2003). Immunoblot analysis of cytosolic and mitochondrial fractions from BAT of AdRiKO and control mice revealed no significant difference in the amount and subcellular localization of HKI and HKII (Fig 5E). However, while mitochondrial hexokinase activity was similar in AdRiKO and control mice, cytosolic hexokinase activity was induced in BAT of cold-exposed control but not AdRiKO mice (Fig 5F and G). Thus, mTORC2 signaling in BAT stimulates glucose uptake and glycolysis upon cold exposure via regulation of cytosolic hexokinase activity. Collectively, the above findings suggest that impaired glucose metabolism in BAT of AdRiKO mice accounts for the failure to sustain NST.

## Restoration of glucose uptake or Akt signaling suppresses the thermogenic defect in AdRiKO mice

Our data suggest that AdRiKO mice are hypothermic and sensitive to cold exposure due to impaired activation of glucose metabolism

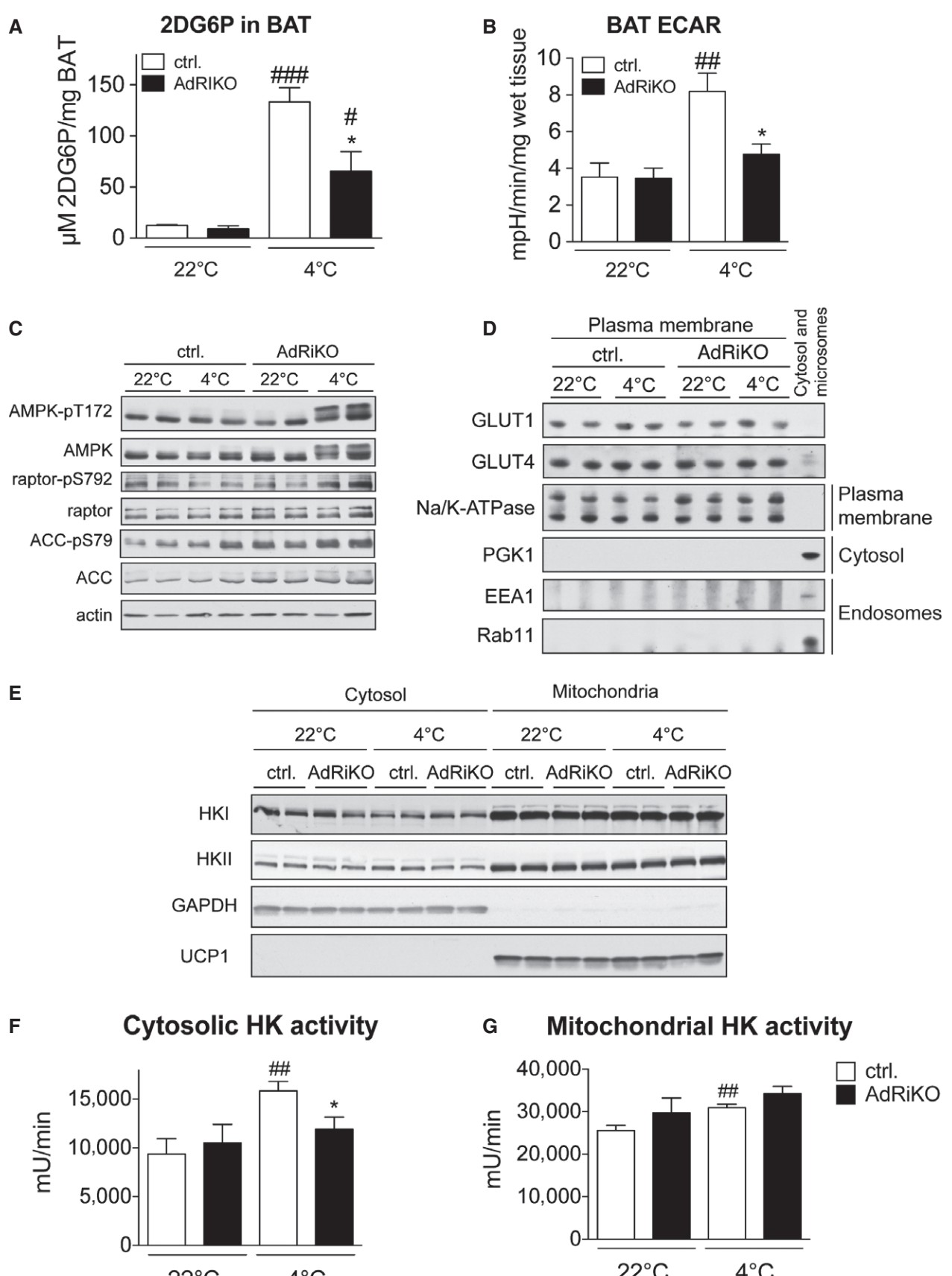

**Figure 5.**

**Figure 5.  mTORC2 in adipose tissue is required for cold-induced glucose uptake and glycolysis.**

A   2-deoxyglucose-6-phosphate (2DG6P) accumulation in BAT of AdRiKO and control mice housed at 22 or at 4°C for 4 h ($n$ = 6/group).
B   Extracellular acidification rate (ECAR) of BAT explants from AdRiKO and control mice housed at 22 or at 4°C for 4 h ($n$ = 7/group).
C   Immunoblot analysis of BAT from AdRiKO and control mice housed at 22 or at 4°C for 8 h for the indicated proteins ($n$ = 6/group, each lane represents a mix of 3 mice).
D   Immunoblot analysis of isolated plasma membranes from BAT of AdRiKO and control mice housed at 22 or at 4°C for 8 h for the indicated proteins ($n$ = 6/group, each lane represents a mix of 3 mice).
E   Immunoblot analysis of mitochondrial and cytosolic fractions from BAT of AdRiKO and control mice housed at 22 or at 4°C for 4 h for the indicated proteins ($n$ = 6/group, each lane represents a mix of 3 mice).
F   Cytosolic hexokinase activity in BAT of AdRiKO and control mice housed at 22 or at 4°C for 4 h [$n$ = 5 (control 22°C), $n$ = 5 (AdRiKO 22°C), $n$ = 7 (control 4°C), $n$ = 7 (AdRiKO 4°C)].
G   Mitochondrial hexokinase activity in BAT of AdRiKO and control mice housed at 22 or at 4°C for 4 h [$n$ = 5 (control 22°C), $n$ = 5 (AdRiKO 22°C), $n$ = 7 (control 4°C), $n$ = 7 (AdRiKO 4°C)].

Data information: Data represent mean ± SEM. Statistically significant differences between AdRiKO and control mice were determined with unpaired Student's $t$-test and are indicated with asterisks (*$P$ < 0.05). Statistically significant differences between temperatures were determined with unpaired Student's $t$-test and are indicated with a number sign (#$P$ < 0.05; ##$P$ < 0.01; ###$P$ < 0.001). The exact $P$-value for each significant difference can be found in Appendix Table S2.

in BAT. We therefore reasoned that restoring glucose uptake in BAT could be sufficient to improve temperature regulation. To test this notion, we overexpressed HKII in BAT of AdRiKO and control mice via intra-BAT injection of an adeno-associated viral vector (AAV) expressing HKII. This technique was used previously to activate glucose uptake specifically in BAT (Jimenez *et al*, 2013). To confirm that the transgene was targeted specifically to BAT, we injected RFP-expressing AAV into the BAT of control mice and measured RFP expression. RFP was strongly expressed in BAT, slightly expressed in liver, and not detected in other tissues (Fig EV4A and B). Intra-BAT injection of an HKII-expressing AAV resulted in a strong increase in HKII mRNA levels in BAT of both AdRiKO and control mice (Fig 6A). Overexpression of HKII enhanced cold-induced glucose uptake in BAT of both AdRiKO and control mice (Fig 6B). Importantly, restoration of glucose uptake suppressed the hypothermia (Fig 6C) and improved cold tolerance (Fig 6D) in AdRiKO mice. Thus, restoration of glucose metabolism in BAT of AdRiKO mice is sufficient to reverse the hypothermia and cold-sensitivity caused by inactivation of mTORC2.

Next we investigated how mTORC2 regulates glucose metabolism and temperature homeostasis in BAT. Akt is a major downstream effector of mTORC2 and stimulates glucose uptake in skeletal muscle and liver (Kumar *et al*, 2008; Hagiwara *et al*, 2012). Furthermore, Akt signaling is activated in BAT upon cold exposure (Fig 2C). Thus, we investigated whether expression of constitutively active Akt2 (Akt2$^{S474D}$) (Hagiwara *et al*, 2012) in BAT of AdRiKO mice could restore glucose uptake and temperature regulation. To this end, we injected AAV expressing Akt2$^{S474D}$ into BAT of AdRiKO and control mice. This resulted in strong expression of Akt2$^{S474D}$ in BAT of AdRiKO and control mice, while PKCα and PKCα-pT638/641 levels were unchanged (Fig 6E). Moreover, Akt2$^{S474D}$ overexpression did not alter plasma insulin levels in AdRiKO or control mice (Fig EV4C). Introduction of Akt2$^{S474D}$ increased body temperature and improved cold tolerance in AdRiKO mice (Fig 6F and G). Thus, restoration of Akt activity in BAT of AdRiKO mice improved temperature homeostasis. To investigate whether restoration of Akt activity also suppressed the observed defects in glucose metabolism, we measured cold-induced glucose uptake in BAT of AdRiKO and control mice. Cold-induced glucose uptake was restored to control levels in BAT of AdRiKO mice expressing Akt2$^{S474D}$ (Fig 6H). In conclusion, these findings suggest that mTORC2-Akt signaling regulates BAT glucose metabolism and thereby NST.

**Figure 6.   Restoration of glucose uptake or Akt signaling suppresses the thermogenic defect in AdRiKO mice.**

A   HKII mRNA expression level in BAT of AdRiKO and control mice infected with either AAV9-HKII or AAV9-empty ($n$ = 8/group).
B   Cold-induced 2-deoxyglucose-6-phosphate (2DG6P) accumulation in BAT of AdRiKO and control mice infected with either AAV9-HKII or AAV9-empty housed at 4°C for 4 h ($n$ = 8/group).
C   Body temperature of AdRiKO and control mice infected with either AAV9-HKII or AAV9-empty housed at 22°C ($n$ = 8/group).
D   Body temperature upon cold exposure of AdRiKO and control mice infected with either AAV9-HKII or AAV9-empty. The left panel represents body temperature after each hour of cold exposure, while the right panel represents body temperature as a bar graph for the 3-h cold exposure time point ($n$ = 8/group). *a*: significant difference between AdRiKO and control mice infected with AAV9-empty; *b*: significant difference between AdRiKO and control mice infected with AAV9-HKII; *d*: significant difference between AdRiKO mice infected with AAV9-empty and AAV9-HKII.
E   Immunoblot analysis of BAT from AdRiKO and control mice infected with either AAV8-Akt2$^{S474D}$ or AAV8-empty ($n$ = 6/group, each lane represents a mix of 3 mice).
F   Body temperature of AdRiKO and control mice infected with either AAV8-Akt2$^{S474D}$ or AAV8-empty housed at 22°C ($n$ = 11/group).
G   Body temperature upon cold exposure of AdRiKO and control mice infected with either AAV8-Akt2$^{S474D}$ or AAV8-empty. The left panel represents body temperature after each hour of cold exposure, while the right panel represents body temperature as a bar graph for the 3-h cold exposure time point ($n$ = 11/group). *a*: significant difference between AdRiKO and control mice infected with AAV8-empty; *b*: significant difference between AdRiKO and control mice infected with AAV8-Akt2$^{S474D}$; *d*: significant difference between AdRiKO mice infected with AAV8-empty and AAV8-Akt2$^{S474D}$.
H   Cold-induced 2-deoxyglucose-6-phosphate (2DG6P) accumulation in BAT of AdRiKO and control mice infected with either AAV8-Akt2$^{S474D}$ or AAV8-empty housed at 4°C for 4 h [$n$ = 7 (control AAV8-null), $n$ = 6 (AdRiKO AAV8-null), $n$ = 6 (control AAV8-Akt$^{S474D}$), $n$ = 6 (AdRiKO AAV8-Akt$^{S474D}$)].

Data information: Data represent mean ± SEM. Statistically significant differences between AdRiKO and control mice were determined with unpaired Student's $t$-test and are indicated with asterisks (*$P$ < 0.05; **$P$ < 0.01). Statistically significant differences between viruses were determined with unpaired Student's $t$-test and are indicated with a number sign (#$P$ < 0.05; ##$P$ < 0.01; ###$P$ < 0.001). The exact $P$-value for each significant difference can be found in Appendix Table S2.

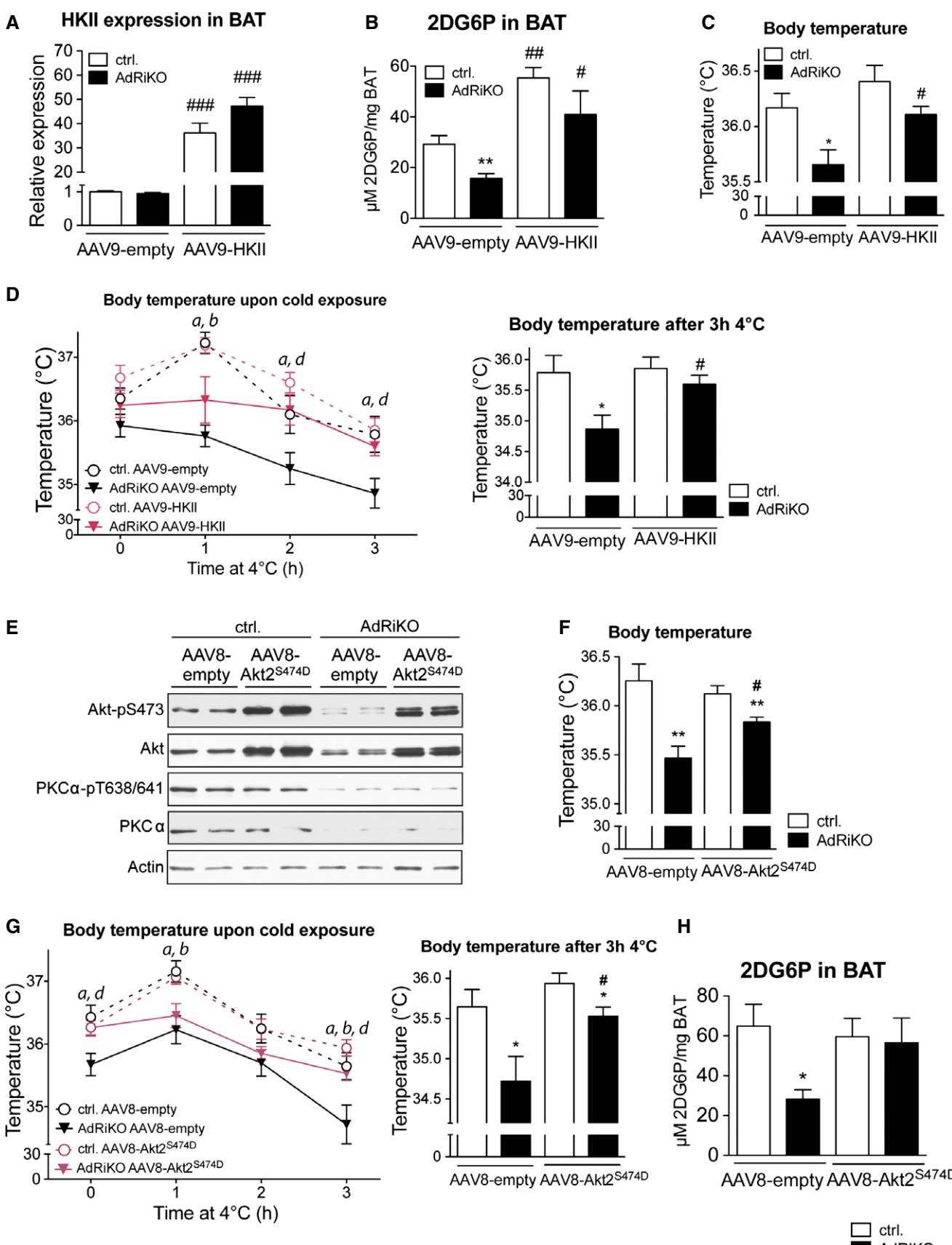

**Figure 6.**

## Discussion

We investigated the role of mTORC2 signaling in the regulation of thermogenesis and cold-induced glucose uptake. We show that mTORC2 signaling is activated in brown adipocytes *in vitro* and *in vivo* upon NE stimulation and cold exposure, via cAMP and Epac1, independently of PKA signaling. We also demonstrate that mTORC2-Akt signaling in BAT mediates cold-induced glucose uptake and glycolysis and thereby sustains NST (Fig 7).

We found that mice with inactive mTORC2 signaling in adipose tissue (AdRiKO mice) fail to maintain a metabolic rate required to sustain NST and are thus hypothermic and sensitive to cold stress. This impaired NST response of AdRiKO mice is most likely due to impaired glucose uptake and glycolysis in BAT, which results in the inability to maintain energy homeostasis under cold stress. Importantly, overexpressing HKII specifically in BAT restored glucose uptake and glycolysis and thereby restored body temperature and improved cold tolerance. These results reveal the importance of glucose metabolism in BAT, and its regulation by mTORC2, in the maintenance of NST. Interestingly, Olsen *et al* reported that β-adrenergic stimulation induced glucose uptake in brown adipocytes *in vitro* in an mTORC2-dependent manner (Olsen *et al*, 2014). Furthermore, Olsen *et al* reported that mTORC2 promotes glucose uptake by stimulating GLUT1 translocation to the plasma membrane in an Akt-independent fashion (Olsen *et al*, 2014). In contrast to these *in vitro* results, we did not observe a change in GLUT1 plasma membrane localization in BAT of AdRiKO mice. Moreover, our *in vivo* results show that mTORC2 mediates cold-stimulated glucose uptake and glycolysis in an Akt-dependent manner. AdRiKO mice displayed a strong decrease in Akt S473 phosphorylation in BAT, and overexpression of a constitutively active Akt2 mutant (Akt2$^{S474D}$) restored glucose uptake and body temperature and increased cold tolerance in AdRiKO mice. The mechanism by which mTORC2 regulates glucose uptake might be different *in vivo* and *in vitro*. How does mTORC2-Akt signaling in BAT stimulate glucose uptake and glycolysis? We found that AdRiKO mice are defective for induction of cytosolic hexokinase activity in BAT upon cold

exposure, whereas mitochondrial hexokinase activity was unaffected. Interestingly, it has been proposed that mitochondria-associated hexokinase utilizes ATP generated by the mitochondria for glucose phosphorylation (Wilson, 2003). However, upon cold stress, mitochondrial ATP production is strongly reduced due to activation of the uncoupling protein UCP1 (Lindberg *et al*, 1967; De Meis *et al*, 2012). Thus, in the context of thermogenesis, cytosolic rather than mitochondrial hexokinase may account for the increase in glucose uptake and glycolytic rate. In summary, our data suggest that mTORC2 in BAT specifically activates cytosolic hexokinase, which is in turn required for cold-induced glucose uptake and glycolysis and maintenance of energy homeostasis upon cold stress. Future studies should address the mechanism by which mTORC2-Akt signaling activates cytosolic hexokinase.

We observed that mTORC2 signaling is activated in brown adipocytes upon adrenergic stimulation. Similar to insulin-induced activation of mTORC2, we found that adrenergic stimulation activates mTORC2 in a PI3K-dependent and mTORC1-independent fashion. Additionally, we found that adrenergic signaling stimulates mTORC2 via cAMP and Epac1, independently of PKA signaling. Interestingly, it has been shown previously that mTORC2 signaling is activated in prostate cancer cells in an Epac1-dependent fashion upon cAMP stimulation (Misra & Pizzo, 2012). Thus, induction of mTORC2 signaling upon adrenergic stimulation seems to occur in several distinct cell types and could thus represent another major input for mTORC2 activation in addition to growth factors. Our results also suggest that mTORC2-Akt signaling in addition to PKA signaling plays an important role in the NST response.

Loss of mTORC2 impaired temperature homeostasis, but without affecting cold-induced β-oxidation, lipid mobilization, and mitochondrial uncoupling. This is in contrast to the study of Hung *et al*, which found that Myf5 muscle- and BAT progenitor cell-specific *rictor* KO (Myf5-*rictor* KO) mice display increased oxidative metabolism and uncoupling in BAT (Hung *et al*, 2014). A possible explanation for these seemingly discrepant results could be that Hung *et al* used a Myf5-driven Cre recombinase to delete *rictor* at the pre-adipocyte stage, whereas we used an aP2-driven Cre recombinase to delete *rictor* only in mature adipocytes. A defect in mTORC2 signaling during adipogenesis could affect mature BAT function and potentially result in changes in oxidative metabolism. Nevertheless, our results demonstrate that inactivation of mTORC2 signaling in mature adipocytes does not affect lipid mobilization, mitochondrial function, or oxidative metabolism in BAT. However, since AdRiKO mice display inactive mTORC2 signaling in both WAT and BAT, some of the observed phenotypes of AdRiKO mice might also be due to defects in WAT. Further studies are required to investigate this possibility.

Taken together, our results demonstrate a novel role for mTORC2 in BAT in the regulation of energy homeostasis and thermogenesis, through Akt-mediated stimulation of glucose uptake and glycolysis (Fig 7). NST and subsequent energy dissipation has been proposed as a novel strategy to treat obesity and decrease the risk of obesity-associated diseases (Clapham & Arch, 2011). Additionally, cold-stimulated glucose uptake could be used to normalize blood glucose levels in insulin-resistant diabetic patients. Our data suggest that activation mTORC2 in BAT, to stimulate glucose metabolism, could have synergistic effects with NST activators in the treatment of obesity.

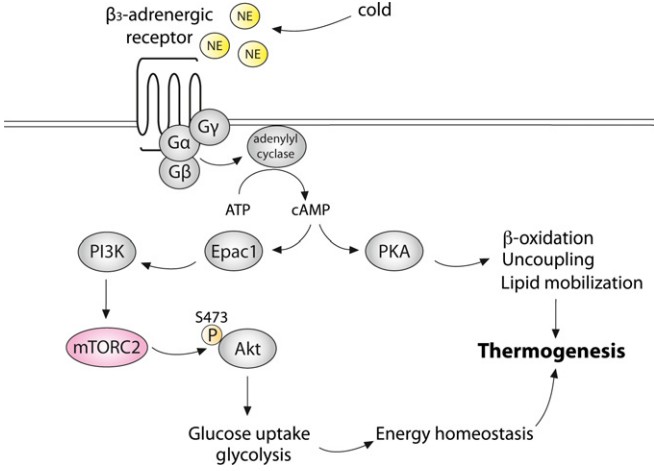

**Figure 7. mTORC2 in BAT is activated by adrenergic stimulation and mediates temperature homeostasis via regulation of cold-induced glucose uptake and glycolysis**.

## Materials and Methods

### Cell culture

SV40T-immortalized C57BL/6 mouse brown pre-adipocytes were kindly provided by Professor Johannes Klein (Lübeck, Germany) (Klein *et al*, 2002). Pre-adipocytes were grown to confluency in Dulbecco's modified Eagle's medium (DMEM; Sigma-Aldrich) supplemented with 20% fetal calf serum (FCS), 4.5 g/l glucose, 2 mM glutamine, 20 nmol/l insulin, and 1 nmol/l triiodothyronine. Twenty-four hours after reaching confluency, adipocyte differentiation was induced by the addition of 250 μmol/l indomethacin, 500 μmol/l isobutylmethylxanthine, and 2 μg/ml dexamethasone to the medium for 24 h. Cell culture was continued for five more days before experiments were performed. All cell culture experiments were performed in three independent replicates, and a representative replicate is presented.

### Animals

Adipose tissue-specific rictor knockout mice (AdRiKO) were already described and characterized previously (Cybulski *et al*, 2009). Mice were housed at 22°C in a conventional facility with a 12-h light/12-h dark cycle. For all experiments, male aP2-Cre; rictor$^{LoxP/LoxP}$ (AdRiKO) and rictor$^{LoxP/LoxP}$ (control) mice between 10 and 14 weeks of age were used. Animals were randomly assigned for measurements or treatments. Experimental groups were formed concerning genotype and similarity in age. Animals that became sick or died during the experiment and those which failed to show a successful experimental performance were excluded from the analysis. All experiments were performed in accordance with the federal guidelines for animal experimentation and were approved by the Kantonales Veterinäramt of Kanton Basel-Stadt.

### *In vivo* norepinephrine stimulation

Mice were starved for 12 h over night prior to norepinephrine administration. Mice were injected intraperitoneally with 1 mg/kg norepinephrine and sacrificed 30 min later.

### Cold exposure

Mice were housed in single cages with free access to water at 4°C for a period of 2, 4, 8 or 10 h. For the 4- and 8-h cold exposure, food was removed specifically during cold exposure. For the 2-h cold exposure, food was removed 12 h prior to cold exposure and during cold exposure to allow assessment of mTORC2 signaling. For the 10-h cold exposure, mice had free access to food and water.

### Thermoneutrality

Mice were housed in single cages with free access to water and food at 30°C for a period of 2 weeks.

### Locomotor activity, metabolic rate, and body temperature

Locomotor activity and metabolic rate was measured in 30-min intervals for the indicated time using a comprehensive laboratory animal monitoring system (CLAMS, Linton Instrumentation and Columbus Instruments) after 24 h of acclimatization. For determination of maximal respiration, the highest measured VO$_2$ value for each mouse was taken. Mice had free access to food and water during the acclimatization period. For the measurement period, food was removed. Body temperature was measured using a rectal thermometer (BAT-12, Physitemp).

### EMG measurement

To measure cold-induced muscle shivering, mice were housed at 4°C for 4 h and subsequently anaesthetized with isoflurane. Then, three 29-gauge needle electrodes (two recording electrodes and one reference electrode placed distally) were placed transcutaneously to acquire the EMG signal from the scapular muscles. For each mouse, the EMG signal was recorded for 5 min. The signal was processed with a low-pass filter of 3 kHz, a high-pass filter of 10 Hz, and a notch filter of 60 Hz. Data were A/D converted and recorded at a sampling frequency of 24 kHz (DantecKeypoint). Root-mean-square (rms) of the EMG signal was calculated.

### *Ex vivo* oxygen consumption and extracellular acidification rate

Oxygen consumption and extracellular acidification rate of BAT were measured using an XF24 extracellular flux analyzer (Seahorse Biosciences). Mice were housed in single cages at either 22 or 4°C for 4 h without food, and subsequently, BAT was collected and cut in approximately 0.5-μg big pieces. Tissues were washed three times with Seahorse assay buffer supplemented with 25 mM glucose, 2 mM glutamine, 1 mM sodium pyruvate adjusted to pH 7.4. Subsequently, BAT pieces were placed in the center of a Seahorse XF24 islet capture microplate containing 675 μl of Seahorse assay buffer. After 30-min incubation at 37°C without CO$_2$, oxygen consumption and extracellular acidification rate were measured 10 times and normalized to tissue weight. From each mouse, five individual BAT pieces were used for measurement.

### BAT triglyceride and free fatty acid measurement

For the free fatty acid measurement, the crude lipid fraction was extracted from BAT of mice housed at either 22 or 4°C for 8 h using choroform:methanol (2:1). From this fraction, the amount of free fatty acids was determined using a commercial kit [HR Series NEFA-HR(2)] and normalized to tissue weight. For triglyceride measurement, the crude lipid fraction was further purified on a solid-phase extraction column (UPTI-CLEAN NH2-S 100 mg/1 ml SPE Colums, Interchim). Subsequently, amount of triglycerides was determined using a commercial kit (TG PAP BioMérieux) and normalized to tissue weight.

### RNA isolation and RT–PCR

Total RNA from BAT of mice housed at either 22 or 4°C for 8 h was isolated with TRIzol reagent (Sigma) and RNeasy kit (Qiagen) followed by cDNA synthesis using iScript cDNA synthesis kit (Bio-Rad). Semiquantitative real-time PCR analysis was performed using fast SYBR green (Applied Biosystems) on a StepOnePlus Real-Time PCR System (Applied Biosystems). Relative expression levels were determined by normalizing to either *RPL0* or *TBP* expression using

the $\Delta\Delta C_T$ method. The sequence for the primers used in this study can be found in Appendix Table S1.

## mtDNA copy number determination

Total DNA was isolated from BAT of mice housed at either 22 or 4°C for 8 h by incubating the tissue in PBND buffer containing 0.1 mg proteinase K at 57°C over night, followed by proteinase K inactivation at 95°C for 10 min. DNA was subsequently purified using a standard chloroform/phenol/isoamyl alcohol precipitation. mtDNA was determined in relation to the genomic DNA by qRT–PCR using primers against the *D-loop* region for mtDNA and against the single-copy nuclear gene *Ndufv1* for genomic DNA. The sequence for the primers used can be found in Appendix Table S1.

## Protein isolation and Western blot

For norepinephrine or 8-Br-cAMP stimulation and subsequent Western blot analysis in cells, cells were starved for 16 h in DMEM supplemented with 1% FCS, 4.5 g/l glucose, and 2 mM glutamine. Cells were pretreated for 30 min with DMSO, 100 nM rapamycin, 125 nM Torin, 100 nM wortmannin, 10 µM ESI-09, or for 2 h with 20 µM H89 and subsequently stimulated with 1 µM of nore-pinephrine or 1 mM 8-Br-cAMP for 5 min unless indicated otherwise. Subsequently, cells were harvested in cold RIPA buffer containing 50 mM Tris–HCl (pH 7.5), 2 mM EDTA, 2 mM EGTA, 150 mM NaCl, 1% NP-40, 0.5% Na-Deoxycholate, 0.1% SDS, protease inhibitors (Roche), and phosphatase inhibitors (Sigma-Aldrich). For protein isolation from adipose tissue, tissue was homogenized in lysis buffer containing 100 mM Tris–HCl (pH 7.5), 2 mM EDTA, 2 mM EGTA, 1% Triton X-100, protease inhibitors (Roche), and phosphatase inhibitors (Sigma-Aldrich). Protein concentration was determined by Bradford assay, and equal amounts of protein were separated on SDS–PAGE followed by transfer onto a nitrocellulose membrane (Whatman). The following antibodies were used to detect the proteins of interest: Akt (Cell Signaling, cs-4685), Akt-pS473 (Cell Signaling, cs-9271), mTOR (Cell Signaling, cs-2972), mTOR-pS2481 (Cell Signaling, cs-2974), Creb (Cell Signaling, cs-9196), Creb-pS133 (Cell Signaling, cs-9197), actin (Millipore, MAB1501), AMPK (Cell Signaling, cs-2532), AMPK-pT172 (Cell Signaling, cs-2535), raptor (Cell Signaling, cs-2280), raptor-pS792 (Cell Signaling, cs-2083), ACC (Cell Signaling, cs-3662), ACC-pS79 (Cell Signaling, cs-3661), perilipin (Cell Signaling, cs-9349), HSL (Cell Signaling, cs-4107), HSL-pS563 (Cell Signaling, cs-4139), PKCα (Cell Signaling, cs-2056), PKCα-pT638/641 (Cell Signaling, cs-9375), Mitoprofile (MitoSciences, MS604), HKI (Cell Signaling, cs-2024), HKII (Cell Signaling, cs-2867), GLUT1 (Abcam, ab-40084), GLUT4 (Novus Biologicals, NBP2-22214), Na/K-ATPase (Cell Signaling, cs-3010), EEA1 (Abcam, ab2900), Rab11 (Millipore, 05-853), UCP1 (Abcam, ab-10983), GAPDH (Cell Signaling, cs-2118), and rictor (Cell Signaling, cs-2114). Antibodies were diluted 1:1,000, except for actin, ACC, and ACC-pS79, which were diluted 1:10,000. Band intensities were quantified using ImageJ software.

## 2-deoxyglucose (2-DG) uptake

For 2-DG uptake measurement in BAT, mice were housed in single cages at 22 or 4°C for 4 h without food. Subsequently, mice were treated with 32.8 µg/kg of 2-DG (Sigma) and sacrificed 45 min later. BAT was harvested and lysed in 10 mM Tris–HCl (pH 8.1). 2-DG uptake was measured by quantifying 2-DG6P accumulation in BAT using a commercial kit (Cosmo Bio Co, LTD.) following the manufacturer's instructions. Values were normalized to tissue weight.

## Isolation of plasma membrane

Plasma membrane and cytosol were isolated from BAT of mice housed at either 22 or 4°C for 8 h by differential centrifugation. For Western blot analysis, the soluble cytosolic and microsomal fraction obtained during plasma membrane isolation was used for comparison. Equal amounts of protein were loaded from each fraction.

## Isolation of crude mitochondria

Crude mitochondrial and cytosolic fractions were isolated from fresh BAT of mice that were housed in single cages at 22 or 4°C for 4 h without food by differential centrifugation.

## Hexokinase activity

Hexokinase activity of crude mitochondrial and cytosolic fractions was determined using a commercial kit (Abcam), following the manufacturer's instructions. Equal amount of protein was used for activity determination.

## Histology and immunostainings

BAT and sWAT of mice housed at either 22 or 4°C for 8 h were fixed over night in 4% paraformaldehyde in PBS at 4°C, dehydrated, embedded in paraffin, and cut into 5-µm-thick sections. Sections were stained with H&E (Merck) to perform general histology. For immunostainings, the following antibody was used: RFP (ab62341, Abcam). DAPI was used to stain nuclei.

## Electron microscopy

BAT of mice housed at either 22 or 4°C for 4 h was fixed in 3% paraformaldehyde/0.5% glutaraldehyde, followed by fixation in 1% osmiumtetroxid and subsequent embedding in epon. Tissue was cut into 60- to 70-nm-thick sections, and images were taken with a Morgagni 268(D) TEM (FEI).

## Blood analysis

Free fatty acids and glycerol in plasma of mice housed at either 22 or 4°C for 8 h were determined using commercial kits (HR Series NEFA-HR(2) and Cayman). Plasma triglycerides were measured using a biochemical analyzer (Cobas c 111 analyzer, Roche). Blood glucose was measured from tail vain using a glucose meter (Accu-check, Roche). Plasma insulin was determined using a commercial kit (Crystal Chem).

## Recombinant AAV vector production and delivery

AAV vector production and delivery into BAT were carried out as described by Jimenez *et al* (Jimenez *et al*, 2013). Briefly,

## The paper explained

**Problem**

Activation of non-shivering thermogenesis (NST) and subsequent dissipation of heat in brown adipose tissue (BAT) has been proposed as an attractive anti-obesity treatment. Additionally, cold-induced glucose uptake could normalize blood glucose levels in insulin-resistant diabetic patients. It is therefore important to identify novel regulators of NST and cold-induced glucose uptake.

**Results**

Here, we show that mammalian target of rapamycin complex 2 (mTORC2) signaling is activated in brown adipocytes upon adrenergic stimulation. Furthermore, we demonstrate that mTORC2 in adipose tissue is essential for temperature homeostasis. Mice lacking mTORC2 specifically in adipose tissue (AdRiKO mice) are hypothermic, display increased sensitivity to cold, and show impaired cold-induced glucose uptake and glycolysis. Restoration of glucose uptake in BAT was sufficient to increase body temperature and improve cold tolerance in AdRiKO mice.

**Impact**

We identified mTORC2 in BAT as a novel regulator of temperature homeostasis via regulation of cold-induced glucose uptake. Our findings demonstrate the importance of glucose metabolism in temperature regulation and could lead to the development of new anti-obesity therapies.

single-stranded AAV vectors were produced through triple transfection of HEK293 cells and subsequent purification on a CsCl gradient. AAV vectors used were as follows: AAV8-CAG-RFP, AAV8-CAG-humanAkt2$^{S474D}$, and AAV9-CMV-HKII. Non-coding plasmids carrying the CAG or CMV promoter and a multicloning site (AAV8-CAG-empty, AAV9-CMV-empty) were used to produce null particles. For intra-BAT AAV administration, mice were anaesthetized with isoflurane and a longitudinal incision was performed at the interscapular area to expose the BAT. Each lobe of the BAT was injected twice with 10 μl of viral solution to distribute the vector in the entire depot. Each mouse received $2 \times 10^{11}$ viral genomes dissolved in 0.001% Pluronics F68 in PBS. Mice were allowed to recover from the surgery for 2 weeks before experiments were performed.

### Data analysis

Sample size was chosen according to our previous studies and published reports in which similar experimental procedures were described. No blinding of investigators was done. All data are expressed as mean ± SEM. To determine statistically significant differences between groups, normal distribution was assumed and unpaired Student's *t*-test was used. *$^{/\#}P < 0.05$, **$^{/\#\#}P < 0.01$, ***$^{/\#\#\#}P < 0.001$.

**Expanded View** for this article is available online.

### Acknowledgements

We would like to thank Professor Brian Hemmings (Basel), Professor Johannes Klein (Lübeck), Professor Stefan Krähenbühl (Basel), and the Imaging Core Facility of the Biozentrum for reagents, equipment, and technical support. We acknowledge support from Ministerio de Economía y Competitividad, Plan Nacional I+D+I (SAF 2014-54866-R), and Generalitat de Catalunya (2014 SGR 1669 and ICREA Academia), Spain (F.B.), and from the Swiss National Science Foundation, the Louis Jeantet Foundation, the Werner Siemens Foundation (VA), and the Canton of Basel (M.N.H.).

### Author contributions

VA, KS and MNH participated in conceptualization and methodology. VA, KS, MS, and MC performed experiments. VA and MNH wrote the manuscript. VA, MNH, and FB contributed funding acquisition. SM, VJ, CH, and FB provided resources. MNH and FB supervised the study.

### Conflict of interest

The authors declare that they have no conflict of interest.

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
