## [Review Process File · EMBO Molecular Medicine]

mTORC2 sustains thermogenesis via Akt-induced glucose uptake and glycolysis in brown adipose tissue

Verena Albert, Kristoffer Svensson, Mitsugu Shimobayashi, Marco Colombi, Sergio Muñoz, Veronica Jimenez, Christoph Handschin, Fatima Bosch, Michael N. Hall

Corresponding author: Michael N. Hall, University of Basel

Review timeline:

Submission date:	09 July 2015
Editorial Decision:	08 August 2015
Revision received:	09 November 2015
Editorial Decision:	08 December 2015
Revision received:	10 December 2015
Accepted:	14 December 2015

Transaction Report:

Editor: Céline Carret

1st Editorial Decision

08 August 2015

Thank you for the submission of your manuscript to EMBO Molecular Medicine. We have now heard back from the three referees whom we asked to evaluate your manuscript.

As you will see from the reports below, the referees find the topic of your study of potential interest. However, they raise substantial concerns on your work, which should be convincingly addressed in a major revision of the present manuscript. The three reports have overlapping concerns and all referees suggest strengthening the conclusiveness of the data, addressing discrepancies found between your study and Olsen et al., characterise better the animals used, improve statistical analyses and reporting. In addition, you will see that referee 1 is not convinced by the cell line work used in Figure 1 and suggests to remove these data rather than fixing it. I would however let that decision to you.

Overall it is clear that publication of the manuscript cannot be considered at this stage. I also note that addressing the reviewers concerns in full will be necessary for further considering the manuscript in our journal and this appears to require a lot of additional work and experimentation. I am unsure whether you will be able or willing to address those and return a revised manuscript within the 3 months deadline. On the other hand, given the potential interest of the findings, I would be willing to consider a revised manuscript with the understanding that the referee concerns must be fully addressed and that acceptance of the manuscript would entail a second round of review.

I should remind you that it is EMBO Molecular Medicine policy to allow a single round of revision only and that, therefore, acceptance or rejection of the manuscript will depend on the completeness of your responses included in the next, final version of the manuscript. For this reason, and to save you from any frustrations in the end I would strongly advise against returning an incomplete revision and would also understand your decision if you chose to rather seek rapid publication elsewhere at this stage.

I look forward to seeing a revised form of your manuscript as soon as possible.

Should you find that the requested revisions are not feasible within the constraints outlined here and choose, therefore, to submit your paper elsewhere, we would welcome a message to this effect.

NB: Please see below important information about formatting and provision of supplementary files

***** Reviewer's comments *****

Referee #1 (Comments on Novelty/Model System):

See my comments on the use of the cell line for Figure 1. Since it is a cell line it is unreliable with respect to signal transduction. It also fails to contribute significantly to the story.

Referee #1 (Remarks):

The manuscript by Albert and colleagues contains interesting information concerning the role of the mTORC2 complex in glucose uptake in brown adipose tissue. The authors also postulate an indirect role for mTORC2 in nonshivering thermogenesis, based on a need for glucose in brown adipose tissue.

The area of glucose uptake in brown adipose tissue (BAT), stimulated primarily by the sympathetic nervous system, was quite extensively studied in the 1980's and early 1990's but has expanded greatly in the last few years because of the use of FDG-PET to detect the presence of active BAT. In itself, this technique demonstrates the very high uptake activity found in the sympathetically stimulated tissue, also in adult humans. The signal transduction pathway of the process has, however, not been equally thoroughly clarified.

Nonetheless, the fact that the sympathetically stimulated uptake is mediated by beta-adrenergic receptors and cyclic AMP is well known. In addition, more recent work (Olsen et al.) has clarified a role for mTORC2 in the glucose uptake pathway in BAT. Thus, the fundamental finding that mTORC2 is involved in sympathetically driven glucose uptake in BAT is not novel. Although there are differences between the results from the primary cell cultures used in Olsen et al. and the animal data reported here, these observations are not at all acknowledged in the Introduction, which consequently needs considerable revision to more fairly acknowledge earlier work in the field.

In addition, there are several other points in need of comment and revision.

1) Fig. 1 concerns studies performed in a BAT-like cell line. Consequently, the value of these studies as reflecting a physiological situation can be questioned. Remarkably, as seen in Fig. 1A, Creb phosphorylation appears to peak first after 2 h of stimulation, in contrast to all earlier work that indicates a peak after about 5 min. In fact, all the responses are markedly slow and thus would appear not to be direct actions of the stimulation. Even the phosphorylation of Akt at Ser473 and Thr450 occurs after 1 to 2 hours. This calls into question the relevance of these studies. Potentially spurious results, such as an effect of H89 on Akt phosphorylation, may further be explainable by H89 acting at one of the other numerous sites it is known to interact with. The evidence that Epac should be involved downstream of mTORC is equally tentative and, without further experimentation, provides little support for the tenet. Similarly, the wortmannin experiment may well not reflect inhibition of PI3K. Indeed, all these studies are performed with single, and in some

cases high, concentrations of inhibitors, which again calls into question their validity.

Some of the data are not in agreement with the study by Olsen and colleagues, but these discrepancies are notably not commented in this section. Since the data from the cell line seem only to have been performed once and lack control experiments for validation of the inhibitors used, I would consequently recommend the authors to delete these cell studies, also because there is no relationship between these and the subsequent studies on glucose uptake and nonshivering thermogenesis, which rather surprisingly was not measured in the cell line.

2) I note that the authors use Akt phosphorylation on Ser473 as indicative of mTORC2 signaling. Although this is a signature activity of mTORC2, it nonetheless seems too limited considering the focus of the article. I was unable to find any data on phosphorylation of the mTORC2 complex and would suggest demonstrating this to provide further evidence for their scheme.

3) The remainder of the article uses a mouse with an adipose tissue-specific ablation of Rictor, generated by the authors some years ago. The mouse has an interesting phenotype, in that it has larger body and organ growth, elevated levels of tissue and circulating IGF1 and IGFBP3 and modest hyperinsulinemia. To what extent these modulations can influence the phenotype observed in the present experiments is not discussed and deserves comment.

Since there are so notable effects of the KO on other tissues and blood-borne factors such as IGF1, a question that arises is whether some of the changes noted can have arisen as a result of an indirect influence on the brain and the sympathetic nervous system. This does not imply that there are not cell-autonomous effects but merely questions whether these measured cell-autonomous effects can fully explain the phenotype. This needs discussion.

4) One aspect of this would be the data in Fig. 2 D,E where the animals defend a lower body temperature, even at 30 {degree sign}C. This would tend to indicate that there is an influence on the body temperature "set-point", which can influence interpretation of other results. Furthermore, have the authors studied the insulation of the KO mouse? Are there any indications of changes in fur or skin quality or in subdermal fat deposition?

5) The observation in Fig. 2F, that the animals are unable to defend body temperature at 4 {degree sign}C, is somewhat strange in that animals without active BAT are able to survive in cold through shivering. Looking at Fig. 2F, I wonder if shivering is functional during the first 3 h, when there is merely a parallel decrease in Tb as in the controls, and only after that do the KOs lose Tb dramatically. Do the muscles become exhausted? Was a loss of shivering observed? Can an inability to shiver for prolonged periods be part of the phenotype?

6) Regarding Fig. 3B, the authors state that there is a comparable increase in NEFAs in plasma following cold exposure. This is clearly not in accordance with the figure and should be commented. There is an inability to elevate NEFAs in the KOs, which may be detrimental for maintenance of shivering.

7) Since the authors show that BAT loses lipid during cold exposure, it is not correct as done in Fig. 4G and H to give oxygen consumption per mg tissue since there will be more active tissue (protein) in the animals that have been exposed to cold.

Also, it is not meaningful to determine oxygen consumption in tissue pieces of BAT in an unstimulated state. Any pre-existing stimulation will have been lost during the preparation. However, addition of norepinephrine is also not meaningful since the tissue pieces quickly become necrotic because oxygen is unable to diffuse into the pieces rapidly enough. I suggest removing these two figures.

8) In Fig. 5A and B, the denominator is again mg tissue and I am concerned that this leads to an overestimate of the uptake values after cold exposure and lipid depletion. Since these are crucial experiments, I would suggest that the authors repeat them and use DNA or protein as denominator.

9) It may be noted that there is apparently no difference in glucose uptake or in rate of extracellular acidification (Fig. 5A,B) between genotypes at 22 {degree sign}C and it is therefore difficult to accept that the inability to maintain normal body temperature at 22 {degree sign}C is related to any inability of BAT to generate heat. Do the authors have any comment (see also point 3 above)? In the same vein, there is a very robust increase in glucose uptake in the KOs in the cold, implying both that there are perhaps other means of stimulating glucose uptake and also that this high level of uptake should rather surprisingly be inadequate. Comments?

10) Indeed, there is a fairly modest increase in hexokinase activity at 4 {degree sign}C in the controls (about 60 %) which is absent in the KOs. However, this activity, seen as a fraction of the total cellular activity, seems minor in view of the dominating activity in the mitochondria. Could the authors please give the relative values? What is the total reduction? Is their conclusion reasonable that this can account for the failure to maintain NST? I find it difficult to be convinced of.

11) The rescue by massive overexpression of hexokinase may indeed be because of enhanced glucose uptake but there is no evidence that this is related to the gene deficit in the KO but may merely circumvent the problem. Glucose uptake is increased more in the controls than in the KOs.

The reason for the requirement for glucose uptake by BAT during thermogenesis is still unclear. It may be, as the authors comment, to maintain cytosolic ATP levels, although the citric acid cycle generates one ATP (GTP) per cycle, so this may not be a main function. Additionally, the mitochondrial uncoupling is certainly not as drastic as that with an artificial uncoupler and ATP generation may occur even in the mitochondria, since the decrease in proton motive force is probably not below the level where this could take place. Glucose metabolism is also required for the anaplerotic reactions needed to maintain fatty acid oxidation, as well as for the generation of glycerol 3-phosphate for lipid synthesis.

12) The final set of experiments regarding overexpression of a constitutively active Akt partly rescues the decrease in body temperature in the KO, both at 22 {degree sign}C and in cold. In addition, glucose uptake is enhanced. The reason for the protection of body temperature may indeed be the enhanced glucose uptake, although there may be other effects and the authors should consider this. I return to the authors' earlier paper on this KO. Although the AAV injections were into BAT, they may e.g. have resolved the hyperinsulinemia and the elevated IGF1 levels.

Minor points

13) Is the subcutaneous depot studied the inguinal? This should be noted. It is of interest in view of recent discussions about the browning of this tissue and its role in systemic glucose uptake.

14) Fig. 4A, the heading should be changed to e.g. Thermogenic genes, as "uncoupling" is clearly misleading.

15) In the same figure, the extent of increase in UCP1 mRNA expression even in the controls is remarkably low. Were the animals provided with large amounts of bedding that could explain the result, in view of the fact that the animals were starved during the experiment? It is difficult to rationalize together with Fig. 2F.

16) The y-axis in Fig. 4H is presumably incorrect.

- 17) The data in Fig. 4I need to be recalculated either to mL per kg lean body mass or per mouse.
- 18) The data in Fig. 4I indeed appear as if the animals successively lose their ability to shiver as the muscle becomes exhausted.
- 19) In Fig. 5C, the sudden appearance of the higher isoform of total AMPK in the KO at 4 {degree sign}C is remarkable and needs comment. In my opinion, it calls into question the conclusion of an activation of AMPK. The ratios of phosphorylated to total do not look notably different. Further experimentation should be done to confirm the authors' conclusion.
- 20) It is perhaps of interest to note that hexokinase activities (Fig. 5F) do not correlate with the relative changes in glucose uptake (Fig. 5A) but do so with the ECAR (Fig. 5B).
- 21) The glucose uptake values in the cold for the controls seem to vary remarkably between experiments - Fig. 5A, 130; Fig. 6B, 30; Fig. 6H, 65. Do the authors have any comment?
- 22) There seems to be notably little GLUT4 in the cytosol, although it is unclear which fraction this refers to and how much protein was used for comparison. This should be clarified. Can the authors demonstrate that insulin indeed induces GLUT4 translocation under these circumstances?

Referee #2 (Remarks):

The paper by Albert et al describes that mTORC2 but not mTORC1 is mandatory for efficient non-shivering thermogenesis during acute cold exposure. On the molecular level the authors show that norepinephrine dependent beta-adrenergic signaling directly stimulate mTORC2 activity thereby mediating glucose uptake and glycolysis. They show that BAT-specific expression of hexokinase or a dominant active form of AKT restore the hypothermic phenotype in mice lacking mTORC2 in adipose tissue. Thus, the authors provide an interesting mechanism how glucose transport is maintained during catabolic conditions of cold exposure. The text is clear and very well written and most of the experiments support the conclusion claimed by the authors. However, I have some concerns that need to be addressed.

1. The authors propose that beta-adrenergic signaling is driving cAMP-dependent glucose uptake in vivo. Insulin has been shown to be very important for efficient glucose uptake into BAT of cold exposed rodents. In addition, it has been shown that free fatty acids induce insulin secretion. As the authors show that free fatty acids are reduced during cold exposure, free fatty induced insulin levels might be reduced in AdRiKO mice. Thus, it is conceivable that reduced insulin levels explain hypothermia of AdRiKO mice during cold exposure. In order to exclude this possibility the authors need to show insulin levels in control and AdRiKO mice during the course of acute cold exposure. In case the authors observe different insulin levels, they should repeat the in vivo glucose uptake studies in control and AdRiKO mice in the presence of the insulin receptor antagonist S961.

2. As the authors mentioned in the discussion, Olsen et al showed that GLUT1 translocation to the plasma membrane induce glucose uptake into brown adipocytes and that this process is dependent on mTORC2 expression. The authors of the current manuscript performed plasma membrane preparations and subsequent Western blot analysis to analyze potential translocation in vivo. In contrast to the results by Olsen, they did not observe an enrichment of GLUT1 and GLUT4 in the plasma membrane fraction suggesting that translocation of glucose transporters to the plasma membrane is unimportant for mTORC2 dependent glucose uptake in vivo. However, plasma membrane preparations of lipid-rich organs such as adipose tissues are often contaminated with endosomal membranes. Thus, the authors need to show that their plasma membrane fractions do not contain endosomal membranes or they have to use alternative methods, e.g. Glut1 as well as Glut4 translocation in response to NE and insulin stimulation should be analyzed in primary brown

adipocytes derived from control and AdRiKO mice.

3. The authors claim that NE-dependent EPAC1 activation induce mTORC2 activation. However, the quality of the respective figure 1D is very poor. The analysis should be repeated.

4. The authors used immortalized brown adipocytes. Signalling pathways might be altered by immortalization. Thus, the authors need to repeat the experiments shown in figure 1B in primary adipocytes isolated from control and AdRiKO mice.

5. Although Fabp4-Cre has been used in many studies to specifically delete floxed genes in adipocytes, in recent years it become clear that Fabp4-Cre expression is detected in the majority but not all adipocytes and its expression is not restricted to adipose tissues. Actually and in order to specifically delete mTORC2 expression in brown adipocytes, it would be necessary to use a transgenic line driving the Cre expression under control of the Ucp1 promotor. This issue should be at least discussed.

Minor points

1. Figures 1B and 1C. Results for Creb are not described in the text.

2. Both PKCa-S638/641 and PKCa-pT638/641 are used in text and figures. Please use the same term.

3. Page 5, last sentence. Since NE is released from the brain upon cold exposure is quite imprecise. Please rephrase the sentence.

Referee #3 (Remarks):

The manuscript by Albert et al. presents the interesting and novel finding that the function of Rictor/mTORC2 in brown adipocytes is required for adaptive thermogenesis and cold tolerance in mice. The authors provide evidence that efficient glucose uptake and glycolysis in brown fat during cold stress depend on Rictor/mTORC2 and that they are necessary for sustained thermogenesis at least under conditions of fasting. In addition, the authors show that mTORC2 is activated by norepinephrine/cAMP (however, this finding is not novel). The authors employ advanced methodology incl. cell type specific knockout as well as rescue through AAV-mediated expression in brown fat. The topic is of great interest for the adipose tissue and metabolic disease fields since brown fat (glucose) metabolism is not well understood yet but relevant for potential therapeutic approaches. The paper has some major weaknesses regarding certain claims as well as some presentation issues which should be feasible to address.

Major points

1. The authors need to strengthen the data supporting the claim that Akt-S473 phosphorylation downstream of NE is specifically dependent on mTORC2. This is important considering that Olsen et al (2014) could not detect increased Akt-S473 phosphorylation in very comparable settings. This point is based on the fact that in vivo (NE or cold in wt/AdRiKO) the Akt-S473-P signal is very variable in the AdRiKO samples (Fig. 2B,C) and has not been quantified (so it is hard to make conclusions on the fold-induction in NE/cold vs. control - there is no doubt that the basal levels depend on Rictor). In the cell culture experiments (Fig. 1), inhibition with Torin/Rapa is not a direct proof for mTORC2 requirement. The authors should either provide quantitation/statistics (min. n=3)

on the *in vivo* Western blot results mentioned above or use knockout or knockdown of Rictor in cultured cells and measure Akt-S473-P upon NE stimulation.

2. The authors have addressed the role of cAMP/Epac/Pka in mTORC2 activation only with 8-Br-cAMP and not with NE. However, they claim that "NE induces mTORC2 signaling via cAMP and PI3K" and "NE activates mTORC2 via cAMP, Epac1" (page 5). This claim cannot be made unless the authors provide data using NE combined with inhibition of at least Epac1 and Pka.
3. The authors should determine blood/serum glucose levels in wt and AdRikO mice after 4-8 hours of cold exposure. This is important for excluding the possibility of systemic differences in glucose metabolism and also for the validity of the tracer experiments.
4. The authors should provide information on the body weights of the mice analyzed as well as the BAT fat pad weights. This is important because many assays incl. O₂ consumption and glucose uptake use normalization to tissue weight.
5. The authors should discuss the fact that the mice were fasted during cold exposure, meaning that mTORC2 function may only be required for thermogenesis under conditions of limiting substrates. Alternatively, the authors should provide experimental data to clarify this point (cold exposure with access to food).
6. The authors should indicate the number of replicates or mice in each figure legend. It is not possible to assess the statistical robustness of the results without the n numbers.

Minor points

7. Literature references on the meaning of the different phosphorylation sites of Akt and their relation to mTORC2 should be provided at latest in the first results section (including a reference proving that the S473 site is a reliable marker for mTORC2 activity).
8. The authors should specify the time point of analysis (duration of treatment/challenge) in the figure legends (Fig 1B-D, Fig. 2A-C, Fig. 3B,C, 4A,B, Fig. 5C-E, Fig. 6B-F).
9. Page 5 last sentence: the authors should correct "NE is released from the brain". NE is released in the periphery by sympathetic nerves (even if the activation of the nerves is initiated in the brain).
10. How was the "maximal respiration" defined and determined? This info is required in the Methods section.
11. What is the difference in the two panels of Fig. 6D? Significance tests should be performed and indicated for the data in the left panel. The same point applies to Fig. 6G.

Referee #1 (Comments on Novelty/Model System):

See my comments on the use of the cell line for Figure 1. Since it is a cell line it is unreliable with respect to signal transduction. It also fails to contribute significantly to the story.

Reply: We have addressed the concerns of Referee #1 regarding the use of a BAT cell line in more detail in the response to comment 1.

Referee #1 (Remarks):

The manuscript by Albert and colleagues contains interesting information concerning the role of the mTORC2 complex in glucose uptake in brown adipose tissue. The authors also postulate an indirect role for mTORC2 in nonshivering thermogenesis, based on a need for glucose in brown adipose tissue.

The area of glucose uptake in brown adipose tissue (BAT), stimulated primarily by the sympathetic nervous system, was quite extensively studied in the 1980's and early 1990's but has expanded greatly in the last few years because of the use of FDG-PET to detect the presence of active BAT. In itself, this technique demonstrates the very high uptake activity found in the sympathetically stimulated tissue, also in adult humans. The signal transduction pathway of the process has, however, not been equally thoroughly clarified.

Nonetheless, the fact that the sympathetically stimulated uptake is mediated by beta-adrenergic receptors and cyclic AMP is well known. In addition, more recent work (Olsen et al.) has clarified a role for mTORC2 in the glucose uptake pathway in BAT. Thus, the fundamental finding that mTORC2 is involved in sympathetically driven glucose uptake in BAT is not novel. Although there are differences between the results from the primary cell cultures used in Olsen et al. and the animal data reported here, these observations are not at all acknowledged in the Introduction, which consequently needs considerable revision to more fairly acknowledge earlier work in the field.

Reply: We thank Referee #1 for the careful evaluation of our manuscript and the insightful suggestions. We acknowledge that Olsen et al. (Olsen et al., 2014) have previously described a role for mTORC2 in glucose uptake in brown adipocytes. The findings described by Olsen et al. are discussed in the "Discussion" part of the manuscript. To accommodate the suggestion from Referee #1, we have now also mentioned the findings by Olsen et al. in more detail in the introduction of the revised manuscript. However, we want to point out that the study by Olsen et al. only investigated the role of mTORC2 for glucose uptake in brown adipocytes *in vitro*. Thus, the study by Olsen et al. does not provide insight into the role of mTORC2-regulated glucose uptake for thermogenesis *in vivo*. In contrast to this, our study provides evidence that mTORC2 in BAT is a major regulator of cold-induced glucose uptake *in vivo* and that this process is crucial for temperature homeostasis. Thus, we believe that our study extends the findings described by Olsen et al. and provides important novel insight into the role of BAT glucose homeostasis for thermogenesis *in vivo*.

In addition, there are several other points in need of comment and revision.

1) Fig. 1 concerns studies performed in a BAT-like cell line. Consequently, the value of these studies as reflecting a physiological situation can be questioned.

Remarkably, as seen in Fig. 1A, Creb phosphorylation appears to peak first after 2 h of stimulation, in contrast to all earlier work that indicates a peak after about 5 min. In fact, all the responses are markedly slow and thus would appear not to be direct actions of the stimulation. Even the phosphorylation of Akt at Ser473 and Thr450 occurs after 1 to 2 hours. This calls into question the relevance of these studies.

Reply: In response to these comments from Referee #1, we have now performed the norepinephrine (NE) stimulation in the BAT cells with a shorter time course. As seen in figure 1A of the revised manuscript, we can confirm that phosphorylation of PKA targets (HSL-pS563 and Creb-pS133) peak 5 minutes after stimulation with NE. Importantly, phosphorylation of Akt-pS473 also peaked after 5 minutes. This suggests that induction of mTORC2 signaling is indeed a direct consequence of NE stimulation and this early induction is not due to indirect effects. These new data can be found in Figure 1A of the revised manuscript. To ensure that our signaling studies were also valid for this early time point, we have repeated the experiments previously presented in Figures 1B, 1C and 1D of the previous version of the manuscript using 5 minutes NE and cAMP stimulation. In line with the previous results obtained with 2 hours NE or 30 minutes of cAMP stimulation, Akt phosphorylation cannot be induced by 5 minutes NE or cAMP stimulation when the cells are treated with Torin, Wortmannin or the Epac1 inhibitor ESI-09. Moreover, treatment of the cells with Rapamycin did not prevent induction of Akt phosphorylation and treatment with H89 resulted in Akt hyperphosphorylation even in the unstimulated state. These new data are presented in Figures 1B-E of the revised manuscript. In conclusion, these novel data suggest that phosphorylation of Akt is a direct consequence of NE and cAMP stimulation and that mTORC2 is activated via cAMP, PI3K and Epac1 but independently of PKA.

Potentially spurious results, such as an effect of H89 on Akt phosphorylation, may further be explainable by H89 acting at one of the other numerous sites it is known to interact with. The evidence that Epac should be involved downstream of mTORC is equally tentative and, without further experimentation, provides little support for the tenet. Similarly, the wortmannin experiment may well not reflect inhibition of PI3K. Indeed, all these studies are performed with single, and in some cases high, concentrations of inhibitors, which again calls into question their validity.

Reply: For the current study, we have used similar concentrations of inhibitors (Rapamycin, Torin, Wortmannin, H89, ESI-09) as previous studies (Almahariq et al., 2013; Holz and Blenis, 2005; Hsieh et al., 2012; Morita et al., 2013; Mullins et al., 2014; Song et al., 2015; Thoreen et al., 2012; Zhang et al., 2006). However, in response to the point made by Referee #1 that we might have used too high concentration of the inhibitors, we have now performed NE stimulation in BAT cells using a 10 times lower concentration of H89 (2 μ M compared to 20 μ M). However, as can be seen in Appended Figure 1A, at this lower concentration of H89, PKA is not fully inhibited and HSL phosphorylation is still induced with NE stimulation. Thus, in the cells used in this study, a higher concentration of H89 must be used to ensure full inhibition of PKA.

Since we observed a PKA-independent induction of mTORC2 signaling with NE or cAMP stimulation, we believe that our findings using the Epac inhibitor provide important insight into the mechanism for how NE activates mTORC2 independently of PKA. Importantly, it has previously been observed that Epac1 mediates cAMP-induced activation of mTORC2/Akt signaling (Misra and Pizzo, 2012). Thus, our results are in accordance with published results and demonstrate that Epac is involved in mTORC2 activation upon β -adrenergic stimulation also in brown adipocytes. We agree with Referee #1 that further experiments would be required to

elucidate the precise mechanism how Epac mediates activation of mTORC2 upon β -adrenergic stimulation. However, at this point we feel that this is beyond the scope of the current study.

Some of the data are not in agreement with the study by Olsen and colleagues, but these discrepancies are notably not commented in this section. Since the data from the cell line seem only to have been performed once and lack control experiments for validation of the inhibitors used, I would consequently recommend the authors to delete these cell studies, also because there is no relationship between these and the subsequent studies on glucose uptake and nonshivering thermogenesis, which rather surprisingly was not measured in the cell line.

Reply: We would like to note that all experiments using the cell line were performed in three individual replicates, and one of these replicates is subsequently shown in the manuscript. We realize that we failed to note this in the previous version of the manuscript, and we apologize for the confusion this might have caused. We have now added this statement in the methods part of the revised version of the manuscript.

Next, regarding the differences between the study by Olsen et al. (Olsen et al., 2014) and our study. One major difference is that the study by Olsen et al. did not observe an induction of Akt phosphorylation with isoproterenol stimulation. Isoproterenol mainly stimulates β -adrenergic receptors, while NE also activates α -adrenergic receptors. Thus, in response to the comments from Referee #1, we hypothesized that the induction of Akt phosphorylation observed with NE stimulation could be due to activation of α -adrenergic receptors, rather than β -adrenergic receptors. To test this notion, we stimulated our cells with isoproterenol. However, in line with our results obtained with NE (Figure 1A) and cAMP (Appended Figure 1B) stimulation, we observed an induction of Akt phosphorylation also with isoproterenol stimulation (Appended Figure 1C). Importantly, this induction was seen after 5 minutes, indicating that this was also a direct effect of isoproterenol stimulation. This finding indicates that β -adrenergic stimulation is sufficient to induce Akt phosphorylation in brown adipocytes. At this point it is difficult to explain the reasons for these observed differences between the Olsen et al. study and our study. However, we would like to stress that we have validated our *in vitro* findings in mice *in vivo*, since NE stimulation also led to activation of Akt phosphorylation in BAT of mice *in vivo* (Figure 2A).

Regarding the relationship between our *in vitro* and *in vivo* studies, we have in this study decided to specifically focus on signaling in the *in vitro* part, and perform subsequent glucose uptake and thermogenesis experiments *in vivo*. However, to answer the question of Referee #1 regarding glucose uptake in BAT cells *in vitro*, we have now performed a glucose uptake assay in these cells. We can observe a reduction in NE-stimulated glucose uptake in cells treated with Torin compared to cells treated with either DMSO or Rapamycin (Appended Figure 1D). Thus these *in vitro* findings are in agreement with our *in vivo* findings of reduced glucose uptake in the absence of mTORC2 in brown adipocytes (Figure 5A).

Finally, we would like to stress that we do not agree with Referee #1 that the *in vitro* data are irrelevant and should be removed from the manuscript. We have several reasons for this: 1) Our new *in vitro* experiments (Figures 1A-1E) suggest that activation of mTORC2 upon NE or cAMP stimulation is a direct effect, since it occurs within 5 minutes. 2) We observe an activation of mTORC2 signaling also in BAT *in*

vivo upon NE stimulation or cold stress, which confirms our *in vitro* findings. 3) Our experiments with NE or cAMP stimulation using different inhibitors (Rapamycin, Torin, Wortmannin, H89, and ESI-09) provide important mechanistic insight into how NE mediates activation of mTORC2. Thus, we believe that the quality of the manuscript would be significantly decreased if we removed the *in vitro* data.

2) I note that the authors use Akt phosphorylation on Ser473 as indicative of mTORC2 signaling. Although this is a signature activity of mTORC2, it nonetheless seems too limited considering the focus of the article. I was unable to find any data on phosphorylation of the mTORC2 complex and would suggest demonstrating this to provide further evidence for their scheme.

Reply: To address this comment, we have now performed Western Blot analysis of mTORC2-pS2481 in BAT of cold-exposed and NE-treated mice. The new data are presented in Figures 2A and 2C of the revised manuscript and demonstrate that mTOR-pS2481 is also induced in BAT of control mice upon NE stimulation or cold exposure.

3) The remainder of the article uses a mouse with an adipose tissue-specific ablation of Rictor, generated by the authors some years ago. The mouse has an interesting phenotype, in that it has larger body and organ growth, elevated levels of tissue and circulating IGF1 and IGFBP3 and modest hyperinsulinemia. To what extent these modulations can influence the phenotype observed in the present experiments is not discussed and deserves comment.

Since there are so notable effects of the KO on other tissues and blood-borne factors such as IGF1, a question that arises is whether some of the changes noted can have arisen as a result of an indirect influence on the brain and the sympathetic nervous system. This does not imply that there are not cell-autonomous effects but merely questions whether these measured cell-autonomous effects can fully explain the phenotype. This needs discussion.

Reply: We would like to point out that the increased body growth, increased organ size and increased IGF-1 levels of AdRiKO mice is only observed upon high fat diet feeding and at a later age (18-20 weeks) (Cybulski et al., 2009). In the present study, we use young (10-14 week old) AdRiKO mice on regular chow diet, which do not display this phenotype. To emphasize this point, we have now included measurements of body composition, body weight and plasma IGF-1 levels of 10-14 week old AdRiKO mice. These data can be found in Figures EV 1B-D of the revised manuscript. At 10-14 weeks of age, AdRiKO mice do not show increased body weight (Figure EV 1B). Moreover, body composition and plasma IGF-1 levels are similar to the levels of control mice (Figures 1C and 1D). In contrast to the unchanged body weight, body composition and IGF-1 levels, young chow-fed AdRiKO mice do however display hyperinsulinemia (Figure EV 3B). Importantly, plasma insulin levels are reduced to a similar extent in both control and AdRiKO mice upon cold exposure (Figure EV 3B). These data are presented in Figure EV 3B of the revised manuscript and indicate that insulin likely plays a minor role in the thermogenic phenotype observed in AdRiKO mice. However, as suggested by Referee #1, we now discuss whether insulin could affect the phenotype of AdRiKO mice in the results section of the revised manuscript.

4) One aspect of this would be the data in Fig. 2 D,E where the animals defend a lower body temperature, even at 30°C. This would tend to indicate that there is an influence on the body temperature "set-point", which can influence interpretation of other results. Furthermore, have the authors studied the insulation of the KO mouse? Are there any indications of changes in fur or skin quality or in subdermal fat deposition?

Reply: The data provided in Figure 2E of the original version of the manuscript were obtained from mice kept at 30°C for only a period of 10 hours. To conclusively answer the question whether the temperature "set-point" of AdRiKO mice is altered, we have now performed a chronic 2 week 30°C exposure. As shown in Figure 2E of the revised manuscript, there is no difference in body temperature between control and AdRiKO mice after 2 weeks chronic 30°C exposure. Thus, AdRiKO mice do not defend a lower body temperature when chronically housed at thermoneutrality. Moreover, we do not observe any obvious changes in fur or skin quality between control and AdRiKO mice (Appended Figure 1E). Finally, AdRiKO mice display a similar amount of subdermal fat as control mice (Figure EV 2B of the revised manuscript). Thus, we conclude that the decrease in body temperature observed in AdRiKO mice does not seem to originate from a change in the body temperature "set-point", fur and skin quality, or subdermal fat deposition.

5) The observation in Fig. 2F, that the animals are unable to defend body temperature at 4 °C, is somewhat strange in that animals without active BAT are able to survive in cold through shivering. Looking at Fig. 2F, I wonder if shivering is functional during the first 3 h, when there is merely a parallel decrease in Tb as in the controls, and only after that do the KOs lose Tb dramatically. Do the muscles become exhausted? Was a loss of shivering observed? Can an inability to shiver for prolonged periods be part of the phenotype?

Reply: To address this question we have now measured cold-induced muscle shivering by electromyography (EMG) after 4h of cold exposure. Unfortunately, we were unable to measure shivering at the later time points of the cold exposure experiment, since at these time points the body temperature of AdRiKO mice was already reduced to the point that the mice did not survive the anesthesia. We therefore measured shivering 4h after cold exposure. At this time point, AdRiKO mice have already lost significantly more body temperature compared to control mice (Figure 2F). Interestingly, we observed that the muscles of AdRiKO mice shivered significantly more after 4h of cold exposure compared to cold-exposed control mice (Figure EV 1J). These data show that shivering in AdRiKO mice is functional up to 4h of cold exposure. Based on these novel findings, we hypothesize that the increased shivering thermogenesis in AdRiKO mice could be a compensatory mechanism to help maintain stable body temperature during cold exposure. At this point we cannot exclude that the muscles of AdRiKO mice become exhausted faster compared to the muscles of control mice due to the enhanced shivering, which could contribute to the rapid loss of body temperature in AdRiKO mice at later time points. However, in such a scenario, we would like to emphasize that the elevated muscle shivering, and the potential early exhaustion of the muscles, is most likely a secondary effect to the loss of thermogenic potential in the BAT of the AdRiKO mice. These new data are presented in Figure EV 1J of the revised version of the manuscript and the potential contribution of muscle shivering to the phenotype of the AdRiKO mice is discussed in the result section of the manuscript.

6) Regarding Fig. 3B, the authors state that there is a comparable increase in NEFAs in plasma following cold exposure. This is clearly not in accordance with the figure and should be commented. There is an inability to elevate NEFAs in the KO mice, which may be detrimental for maintenance of shivering.

Reply: We agree with Referee #1 that AdRiKO mice display significantly decreased levels (-25%) of NEFAs in plasma compared to control mice following cold exposure (Figure 3B). However, AdRiKO mice are still able to significantly increase the amount of circulating NEFAs (+180%) upon cold exposure compared to AdRiKO mice housed at 22°C. Since the levels of circulating NEFAs are approximately 2-fold higher in cold-exposed AdRiKO mice compared to AdRiKO mice kept at 22°C (Figure 3B), we consider that AdRiKO mice do not display an inability to elevate NEFAs in the circulation upon cold exposure. Additionally, lower circulating levels of NEFAs in AdRiKO mice could also reflect changes in NEFA uptake into BAT, since NEFA levels in BAT are significantly higher in AdRiKO mice upon cold exposure (Figure 3F). This could explain why plasma NEFA levels are slightly lower in plasma of cold-exposed AdRiKO mice. However, we would like to emphasize that we agree with Referee #1 that the statement “there is a comparable increase in NEFAs in plasma following cold exposure” is misleading, and we have rephrased this sentence in the results section of the revised manuscript.

7) Since the authors show that BAT loses lipid during cold exposure, it is not correct as done in Fig. 4G and H to give oxygen consumption per mg tissue since there will be more active tissue (protein) in the animals that have been exposed to cold.

Reply: We believe that Referee #1 refers to Figure 3D of the manuscript for this particular point. In the manuscript we state “AdRiKO mice housed at 22°C displayed larger lipid droplets compared to control mice. At 4°C lipid droplet size in BAT decreased to the same extent in AdRiKO and control mice.” In this case, we only refer to the lipid droplet size, and not to the absolute amount of triglycerides present in the tissue. To be able to make a conclusive statement regarding BAT triglyceride content, we have now measured the amount of triglycerides in BAT of AdRiKO and control mice housed at either 22°C or 4°C. These data are presented in Figure 3E, and clearly demonstrate that the amount of triglycerides in the tissue does not change during cold exposure or between genotypes. Moreover, absolute BAT fat pad weight is also unchanged during cold exposure and between genotypes (Figure EV 2A). We believe that these novel findings suggest that normalization with tissue weight is a good way to normalize our data. We have now added the triglyceride quantification to the manuscript, and clarified that while lipid droplet size seems to change, absolute BAT triglyceride content does not change.

Also, it is not meaningful to determine oxygen consumption in tissue pieces of BAT in an unstimulated state. Any pre-existing stimulation will have been lost during the preparation. However, addition of norepinephrine is also not meaningful since the tissue pieces quickly become necrotic because oxygen is unable to diffuse into the pieces rapidly enough. I suggest removing these two figures.

Reply: We agree with Referee #1 that determination of oxygen consumption from tissue *ex vivo* is challenging. The method we implemented was used in other studies to specifically determine oxygen consumption of BAT (Gerhart-Hines et al., 2013; Vergnes et al., 2011). To keep necrosis as low as possible we have cut the tissue in very small pieces (0.5 mg) and only measured one animal per condition

simultaneously. As our data show, we can robustly measure increased *ex vivo* oxygen consumption of BAT tissue pieces and we therefore believe that our data on BAT oxygen consumption is valid and an important piece of data to show.

8) In Fig. 5A and B, the denominator is again mg tissue and I am concerned that this leads to an overestimate of the uptake values after cold exposure and lipid depletion. Since these are crucial experiments, I would suggest that the authors repeat them and use DNA or protein as denominator.

Reply: As mentioned in response to comment number 6, we do not observe any differences in BAT weight or BAT triglyceride content (Figure EV 2A and Figure 3E). Thus, we believe that the normalization with tissue weight is a good way to normalize our data.

9) It may be noted that there is apparently no difference in glucose uptake or in rate of extracellular acidification (Fig. 5A,B) between genotypes at 22 °C and it is therefore difficult to accept that the inability to maintain normal body temperature at 22 °C is related to any inability of BAT to generate heat. Do the authors have any comment (see also point 3 above)? In the same vein, there is a very robust increase in glucose uptake in the KOs in the cold, implying both that there are perhaps other means of stimulating glucose uptake and also that this high level of uptake should rather surprisingly be inadequate. Comments?

Reply: We agree with Referee #1 that glucose uptake does not seem to be impaired in BAT of AdRiKO mice when kept at 22°C (Figure 5A). However, when rescuing glucose metabolism with HKII or Akt2^{S474D} over-expression, we are able to improve both cold tolerance and increase the body temperature of AdRiKO mice when housed at 22°C (Figures 6C and 6F). This suggests that glucose metabolism in BAT might indeed be involved in regulating body temperature at 22°C in AdRiKO mice. It might however be, that the differences at 22°C are too small to be detected by our measurement methods. We agree with Referee #1 that there is an increase in glucose uptake in AdRiKO mice with cold exposure, which indeed suggests that there might be several mechanisms to stimulate glucose uptake into BAT. We feel that it is difficult to judge how much glucose uptake into BAT is required to sustain thermogenesis. Our data show that AdRiKO mice are unable to maintain a stable body temperature (Figure 2F), despite the increase in glucose uptake in BAT of AdRiKO mice with cold exposure (Figure 5A). However, when we enhance glucose uptake (with HKII or Akt2^{S474D} over-expression) to control levels or higher, AdRiKO mice are more cold tolerant (Figures 6D and 6G). These data suggest that the basal increase in glucose uptake in AdRiKO mice (Figure 5A) might not be sufficient to maintain body temperature.

10) Indeed, there is a fairly modest increase in hexokinase activity at 4 °C in the controls (about 60 %) which is absent in the KOs. However, this activity, seen as a fraction of the total cellular activity, seems minor in view of the dominating activity in the mitochondria. Could the authors please give the relative values? What is the total reduction? Is their conclusion reasonable that this can account for the failure to maintain NST? I find it difficult to be convinced of.

Reply: When measuring the relative Hexokinase activities of the cytosolic and mitochondrial fractions, we can indeed measure higher Hexokinase activity in the mitochondrial compartment compared to the cytosol (Figure 5F and 5G). However,

we hypothesize that during cold exposure *in vivo*, mitochondria-generated ATP (which is used by mitochondrial Hexokinase) becomes limiting. Thus, under these conditions we would suggest that cytosolic Hexokinase is more important for cold-induced glycolysis. In contrast, when we determine Hexokinase activity, exogenous ATP is provided for the reaction (see Kit from Abcam, ab136957). Thus, in the assay the limiting amounts of ATP at the mitochondria cannot be mimicked and an increased relative activity of the mitochondrial fraction will be measured. However, this will most likely not correspond to the actual *in vivo* Hexokinase activity. Given the fact that we detect decreased glucose uptake and decreased ECAR without changes in glucose transporters in BAT of AdRiKO mice, we believe that a decrease in cytosolic Hexokinase activity could indeed account for the impaired glucose metabolism observed in BAT of AdRiKO mice. Nevertheless, following the Referee's suggestion we now show the relative Hexokinase activities in the Figure displayed in the manuscript (Figures 5F and 5G).

11) The rescue by massive overexpression of hexokinase may indeed be because of enhanced glucose uptake but there is no evidence that this is related to the gene deficit in the KO but may merely circumvent the problem. Glucose uptake is increased more in the controls than in the KOs.

The reason for the requirement for glucose uptake by BAT during thermogenesis is still unclear. It may be, as the authors comment, to maintain cytosolic ATP levels, although the citric acid cycle generates one ATP (GTP) per cycle, so this may not be a main function. Additionally, the mitochondrial uncoupling is certainly not as drastic as that with an artificial uncoupler and ATP generation may occur even in the mitochondria, since the decrease in proton motive force is probably not below the level where this could take place. Glucose metabolism is also required for the anaplerotic reactions needed to maintain fatty acid oxidation, as well as for the generation of glycerol 3-phosphate for lipid synthesis.

Reply: We agree with Referee #1 that glucose might also be used for other reactions than to maintain cytosolic ATP levels upon cold exposure. We have now mentioned these additional metabolic pathways in the results sections of the revised version of the manuscript.

12) The final set of experiments regarding overexpression of a constitutively active Akt partly rescues the decrease in body temperature in the KO, both at 22 °C and in cold. In addition, glucose uptake is enhanced. The reason for the protection of body temperature may indeed be the enhanced glucose uptake, although there may be other effects and the authors should consider this. I return to the authors' earlier paper on this KO. Although the AAV injections were into BAT, they may e.g. have resolved the hyperinsulinemia and the elevated IGF1 levels.

Reply: As mentioned in the answer to comment 3, we do not observe an increase in circulating IGF-1 levels in 10-14 weeks old AdRiKO mice. Thus, it is unlikely that IGF-1 is involved in the temperature phenotype observed in AdRiKO mice. To address whether constitutively active Akt has an effect on insulin we have now measured plasma insulin in cold-exposed AdRiKO and control mice infected with either AAV8-empty or AAV8-Akt2^{S474D} (Figure EV 4C). Overexpression of a constitutively active version of Akt2 did not significantly alter plasma insulin levels in either control or AdRiKO mice. Thus, we believe that the reason for the protection of body temperature in AdRiKO mice infected with AAV8-Akt2^{S474D} is not due to changes in circulating insulin levels.

Minor points

13) Is the subcutaneous depot studied the inguinal? This should be noted. It is of interest in view of recent discussions about the browning of this tissue and its role in systemic glucose uptake.

Reply: We have indeed studied the inguinal subcutaneous white adipose tissue. We have now noted this in the revised version of the manuscript.

14) Fig. 4A, the heading should be changed to e.g. Thermogenic genes, as "uncoupling" is clearly misleading.

Reply: We have now changed the heading of Figure 4A to "thermogenic genes".

15) In the same figure, the extent of increase in UCP1 mRNA expression even in the controls is remarkably low. Were the animals provided with large amounts of bedding that could explain the result, in view of the fact that the animals were starved during the experiment? It is difficult to rationalize together with Fig. 2F.

Reply: We agree with Referee #1 that the increase in UCP1 mRNA in our experiment is rather low. At this point we do not have any explanation for this, since we did not provide any bedding during the cold exposure. However, we would like to point out that induction of UCP1 mRNA expression during acute cold exposure does not result in an increase in UCP1 protein levels. Consequently, the increase in uncoupling and oxygen consumption upon acute cold stress results from an increased activity of already existing UCP1 protein (Nedergaard and Cannon, 2013). Importantly, we cannot find any decrease in UCP1 protein levels in BAT of AdRiKO mice. This suggests that the inability to sustain a stable body temperature upon cold stress observed in AdRiKO mice most likely does not result from impaired UCP1 protein content and activity.

16) The y-axis in Fig. 4H is presumably incorrect.

Reply: The y-axis in Figure 4H is correct. As explained in the response to comment 10 of Referee #3, for determination of maximal respiration the highest measured VO_2 value during the experiment for each mouse was used. This value is thus still represented as mL/kg/min.

17) The data in Fig. 4I need to be recalculated either to mL per kg lean body mass or per mouse.

Reply: As mentioned in the response to point 3, we do not observe any differences in either body weight or body composition between control and AdRiKO mice. Thus, we consider that the normalization to body weight is a good way to normalize our data.

18) The data in Fig. 4I indeed appear as if the animals successively lose their ability to shiver as the muscle becomes exhausted.

Reply: As mentioned in the answer to point 5 of Referee #1, we measured cold-induced muscle shivering in AdRiKO and control mice by EMG. We observed that the muscles of AdRiKO mice shivered significantly more after 4h of 4°C exposure (Figure EV 1J). At this point we cannot exclude that the muscles of AdRiKO mice become exhausted faster compared to the muscles of control mice due to the enhanced shivering (see reply to point 5 of Referee #1). However, in such a scenario, we would like to emphasize that the elevated muscle shivering, and the potential early exhaustion of the muscles, is most likely a secondary effect to the loss of thermogenic potential in the BAT of the AdRiKO mice.

19) In Fig. 5C, the sudden appearance of the higher isoform of total AMPK in the KO at 4 °C is remarkable and needs comment. In my opinion, it calls into question the conclusion of an activation of AMPK. The ratios of phosphorylated to total do not look notably different. Further experimentation should be done to confirm the authors' conclusion.

Reply: To address this point we have performed and quantified Western Blots for raptor-pS792 and ACC-pS79 from 6 individual mice of each condition used for the Western Blot in Figure 5C (Figure EV 3C). The quantification shows that AdRiKO mice indeed display increased phosphorylation of the AMPK targets raptor and ACC. We agree with Referee #1 that the appearance of a higher total AMPK in AdRiKO mice at 4°C is remarkable. At this point we do not know which post-translational modification could account for this. However, as suggested by the Referee, we have now mentioned the appearance of this isoform in the results part of the revised manuscript. However, due to the clear induction of raptor and ACC phosphorylation in cold-exposed AdRiKO mice, we believe that we can conclude that we have a higher activation of AMPK in BAT of cold-exposed AdRiKO mice compared to cold-exposed control mice.

20) It is perhaps of interest to note that hexokinase activities (Fig. 5F) do not correlate with the relative changes in glucose uptake (Fig. 5A) but do so with the ECAR (Fig. 5B).

Reply: It is indeed an interesting observation and we thank Referee #1 for pointing this out to us. This likely indicates that ECAR is mainly dependent on hexokinase activity while glucose uptake is not. Moreover, this also suggests – as pointed out by Referee #1 in point 9 – that there are probably also other means of stimulating glucose uptake in BAT besides changes in hexokinase activity.

21) The glucose uptake values in the cold for the controls seem to vary remarkably between experiments - Fig. 5A, 130; Fig. 6B, 30; Fig. 6H, 65. Do the authors have any comment?

Reply: We agree with Referee #1 that the glucose uptake values vary between experiments. At this point we can only speculate about possible explanations for these variations. Since we are using an enzyme-based assay this might result from slight differences between different kit batches and from slight differences in incubation times between different experiments. However, we want to point out that

we performed the assay for all samples of one experiment at the same time with the same kit. While the absolute values vary from assay to assay we always consistently observe a decrease in cold-induced glucose uptake in BAT of AdRiKO mice.

22) There seems to be notably little GLUT4 in the cytosol, although it is unclear which fraction this refers to and how much protein was used for comparison. This should be clarified. Can the authors demonstrate that insulin indeed induces GLUT4 translocation under these circumstances?

Reply: The “cytosol” shown in Figure 5D represents the soluble cytosolic and microsomal fraction obtained during the isolation of the plasma membrane by differential centrifugation. We have therefore changed the labeling in Figure 5D to cytosol and microsomes. For Western Blot analysis, we loaded equal amounts of protein (20ug) of both fractions. This might explain why the levels of GLUT4 are very low in the cytosolic/microsomal fraction compared to the plasma membrane fraction. We have now clarified this issue in the methods part of the revised version of the manuscript. Moreover, to demonstrate that our plasma membrane fraction is not contaminated with endosomes, we have now - as suggested by Referee #2 - included Western Blot analysis for the endosomal markers EEA1 and Rab11 (Figure 5D). This analysis clearly shows that endosomal markers are absent in the plasma membrane fractions. To investigate whether we could observe GLUT4 translocation with insulin stimulation, we injected fasted control mice with vehicle or 0.75 U/kg Insulin and determined GLUT4 levels at the plasma membrane. As can be seen in Appended Figure 2A, we can indeed observe an enrichment of GLUT4 at the plasma membrane with insulin stimulation.

Referee #2 (Remarks):

The paper by Albert et al describes that mTORC2 but not mTORC1 is mandatory for efficient non-shivering thermogenesis during acute cold exposure. On the molecular level the authors show that norepinephrine dependent beta-adrenergic signaling directly stimulate mTORC2 activity thereby mediating glucose uptake and glycolysis. They show that BAT-specific expression of hexokinase or a dominant active form of AKT restore the hypothermic phenotype in mice lacking mTORC2 in adipose tissue. Thus, the authors provide an interesting mechanism how glucose transport is maintained during catabolic conditions of cold exposure. The text is clear and very well written and most of the experiments support the conclusion claimed by the authors. However, I have some concerns that need to be addressed.

We would like to thank Referee #2 for the thorough evaluation of our manuscript and the helpful suggestions. Please see below for the specific point-by-point reply to all the concerns raised.

1. The authors propose that beta-adrenergic signaling is driving cAMP-dependent glucose uptake in vivo. Insulin has been shown to be very important for efficient glucose uptake into BAT of cold exposed rodents. In addition, it has been shown that free fatty acids induce insulin secretion. As the authors show that free fatty acids are reduced during cold exposure, free fatty induced insulin levels might be reduced in AdRiKO mice. Thus, it is conceivable that reduced insulin levels explain hypothermia of AdRiKO mice during cold exposure. In order to exclude this possibility the authors need to show insulin levels in control and AdRiKO mice during the course of acute cold exposure. In case the authors observe different insulin levels, they should repeat the in vivo glucose uptake studies in control and AdRiKO mice in the presence of the insulin receptor antagonist S961.

Reply: To answer this point raised by Referee #2, we have now measured plasma insulin levels in our mice at room temperature and during cold exposure. As previously shown (Cybulski et al., 2009), AdRiKO mice display hyperinsulinemia at room temperature (Figure EV 3B). However, during cold exposure, circulating insulin levels are decreased to the same low absolute value in both control and AdRiKO mice (Figure EV 3B). This finding is in line with a study by Gasparetti et al. (Gasparetti et al., 2003) which showed that circulating insulin levels are decreased in mice during cold exposure. This suggests that insulin plays only a minor role in cold-induced glucose uptake. Additionally, since insulin levels were reduced to a comparable extent in control and AdRiKO mice upon cold exposure, we conclude that it is unlikely that the defect in cold-induced glucose uptake observed in AdRiKO mice is mediated by changes in circulating insulin levels. Thus, we decided not to perform any further experiments using the insulin receptor antagonist S961.

2. As the authors mentioned in the discussion, Olsen et al showed that GLUT1 translocation to the plasma membrane induce glucose uptake into brown adipocytes and that this process is dependent on mTORC2 expression. The authors of the current manuscript performed plasma membrane preparations and subsequent Western blot analysis to analyze potential translocation in vivo. In contrast to the results by Olsen, they did not observe an enrichment of GLUT1 and GLUT4 in the plasma membrane fraction suggesting that translocation of glucose transporters to the plasma membrane is unimportant for mTORC2 dependent glucose uptake in vivo. However, plasma membrane preparations of lipid-rich organs such as adipose tissues are often contaminated with endosomal membranes. Thus, the authors need

to show that their plasma membrane fractions do not contain endosomal membranes or they have to use alternative methods, e.g. Glut1 as well as Glut4 translocation in response to NE and insulin stimulation should be analyzed in primary brown adipocytes derived from control and AdRiKO mice.

Reply: To address this question we have now performed Western Blot analysis of the plasma membrane and cytosolic/microsome fractions for the endosomal markers Rab11 and EEA1. As can be seen in Figure 5D of the revised manuscript, we do not detect Rab11 or EEA1 in the plasma membrane fractions. This suggests that the plasma membrane preparations are not contaminated with endosomal membranes.

3. The authors claim that NE-dependent EPAC1 activation induce mTORC2 activation. However, the quality of the respective figure 1D is very poor. The analysis should be repeated.

Reply: In response to the Referee's comment, we have now repeated the experiment presented in Figure 1D of the previous version of the manuscript. We believe that the new data more clearly demonstrates that Akt phosphorylation cannot be induced by cAMP stimulation when Epac1 is inhibited using ESI-09 (Figure 1E). Moreover, following the suggestion of Referee #3, we have also performed NE stimulation in combination with ESI-09 treatment (Figure 1D). ESI-09 treatment also prevented induction of mTORC2 signaling induced by NE stimulation. These new data are shown in Figure 1D and 1E of the revised manuscript.

4. The authors used immortalized brown adipocytes. Signalling pathways might be altered by immortalization. Thus, the authors need to repeat the experiments shown in figure 1B in primary adipocytes isolated from control and AdRiKO mice.

Reply: We agree with Referee #2 that signaling pathways might be altered in immortalized cell lines compared to primary cells. As Referee #2 suggested, we isolated primary brown adipocytes from control and AdRiKO mice. Unfortunately, while control brown adipocytes were able to differentiate, brown adipocytes isolated from AdRiKO mice were unable to differentiate *in vitro* (Appended Figure 2B). This is most likely due to the requirement of insulin, and thus mTORC2 signaling, for brown adipocyte differentiation *in vitro*, as was already shown by Hung et al. (Hung et al., 2014). We are therefore unable to address this point due to technical reasons, and we apologize for this.

5. Although Fabp4-Cre has been used in many studies to specifically delete floxed genes in adipocytes, in recent years it become clear that Fabp4-Cre expression is detected in the majority but not all adipocytes and its expression is not restricted to adipose tissues. Actually and in order to specifically delete mTORC2 expression in brown adipocytes, it would be necessary to use a transgenic line driving the Cre expression under control of the Ucp1 promotor. This issue should be at least discussed.

Reply: We agree with Referee #2 that the Fabp4-Cre line does not exclusively target brown adipocytes and that the use of a UCP1-Cre line would be more specific. However, as shown in our previous publication (Cybulski et al., 2009), we would like to point out that we did not observe reduced rictor levels in any non-adipose organ tested (liver, heart, kidney, spleen, brain, and macrophages). Nevertheless, some of the observed effects on temperature regulation could potentially be due to the

deletion of rictor in white adipocytes. We have now discussed this possibility in the discussion part of the revised manuscript.

Minor points

1. Figures 1B and 1C. Results for Creb are not described in the text.

Reply: First, we would like to point out that we repeated the cell experiments with a shorter stimulation time (in response to the comments from Referee #1). For these experiments we used HSL S563 phosphorylation (instead of Creb S133 phosphorylation) as a readout for PKA activation. While both Creb and HSL are PKA targets, Creb S133 can also be phosphorylated by other kinases, such as p90RSK, MSK, CaMKIV, and MAPKAPK-2 (Ribar et al., 2000; Tan et al., 1996; Xing et al., 1998). In contrast to this, PKA is so far the only identified kinase that phosphorylates HSL at S563 (Garton et al., 1988). Thus, we believe that using HSL S563 is a more accurate readout for PKA signaling activation. In response to the comment from Referee #2, we have now included the description of PKA signaling (HSL S563 phosphorylation) for Figures 1B-1E in the results section of the revised version of the manuscript.

2. Both PKCa-S638/641 and PKCa-pT638/641 are used in text and figures. Please use the same term.

Reply: We thank Referee #2 for pointing out this mistake. We have now changed PKCa-S638/641 to PKCa-pT638/641 throughout the manuscript.

3. Page 5, last sentence. Since NE is released from the brain upon cold exposure is quite imprecise. Please rephrase the sentence.

Reply: We have now rephrased the sentence to “NE is released from sympathetic nerves”.

Referee #3 (Remarks):

The manuscript by Albert et al. presents the interesting and novel finding that the function of Rictor/mTORC2 in brown adipocytes is required for adaptive thermogenesis and cold tolerance in mice. The authors provide evidence that efficient glucose uptake and glycolysis in brown fat during cold stress depend on Rictor/mTORC2 and that they are necessary for sustained thermogenesis at least under conditions of fasting. In addition, the authors show that mTORC2 is activated by norepinephrine/cAMP (however, this finding is not novel). The authors employ advanced methodology incl. cell type specific knockout as well as rescue through AAV-mediated expression in brown fat. The topic is of great interest for the adipose tissue and metabolic disease fields since brown fat (glucose) metabolism is not well understood yet but relevant for potential therapeutic approaches. The paper has some major weaknesses regarding certain claims as well as some presentation issues which should be feasible to address.

We thank Referee #3 for the review of our manuscript and the insightful suggestions. Please see below for the specific point-by-point answers to all the concerns raised.

Major points

1. The authors need to strengthen the data supporting the claim that Akt-S473 phosphorylation downstream of NE is specifically dependent on mTORC2. This is important considering that Olsen et al (2014) could not detect increased Akt-S473 phosphorylation in very comparable settings. This point is based on the fact that in vivo (NE or cold in wt/AdRiKO) the Akt-S473-P signal is very variable in the AdRiKO samples (Fig. 2B,C) and has not been quantified (so it is hard to make conclusions on the fold-induction in NE/cold vs. control - there is no doubt that the basal levels depend on Rictor). In the cell culture experiments (Fig. 1), inhibition with Torin/Rapa is not a direct proof for mTORC2 requirement. The authors should either provide quantitation/statistics (min. n=3) on the in vivo Western blot results mentioned above or use knockout or knockdown of Rictor in cultured cells and measure Akt-S473-P upon NE stimulation.

Reply: In response to the comments of Referee #3, we have now quantified the Akt-pS473 signal of the Western Blots presented in Figures 2B and 2C. To quantify cold-induced Akt-pS473 (Figure 2C) with n≥3, we repeated the Western Blot for Akt and Akt-pS473 with 6 mice for each condition (the same lysates as used in Figure 2C). This Western Blot is shown in Appended Figure 2C. For the quantification of NE-induced Akt-pS473 (Figure 2B) we have quantified the Western Blot presented in Figure 2B (n=3). These quantifications are presented in Figures EV 1E and 1F of the revised version of the manuscript. As evident from the quantifications, both NE stimulation and cold exposure significantly increase Akt-pS473 in BAT of control mice. However, in both cases, Akt S473 phosphorylation is significantly lower in BAT of AdRiKO mice. Since the quantifications clearly demonstrate that Akt phosphorylation is induced by cold and NE only in control but not in AdRiKO mice we decided not to pursue further experiments in cell culture.

2. The authors have addressed the role of cAMP/Epac/Pka in mTORC2 activation only with 8-Br-cAMP and not with NE. However, they claim that "NE induces mTORC2 signaling via cAMP and PI3K" and "NE activates mTORC2 via cAMP, Epac1" (page 5). This claim cannot be made unless the authors provide data using NE combined with inhibition of at least Epac1 and Pka.

Reply: To address this point, we have now performed NE stimulation combined with Epac1 and PKA inhibition in BAT cells (Figure 1D). In line with the result obtained with cAMP stimulation, we observe that Akt phosphorylation cannot be induced with NE stimulation when Epac1 is inhibited. Moreover, inhibition of PKA using H89 again led to hyperphosphorylation of Akt, which could not be further stimulated with NE treatment. These new data demonstrate that NE activates mTORC2 via cAMP and Epac1 and are presented in Figure 1D of the revised manuscript.

3. The authors should determine blood/serum glucose levels in wt and AdRiKO mice after 4-8 hours of cold exposure. This is important for excluding the possibility of systemic differences in glucose metabolism and also for the validity of the tracer experiments.

Reply: In response to this point, we have now measured blood glucose levels after 8h of cold exposure in AdRiKO and control mice. As shown in Figure EV 3A of the revised manuscript, blood glucose levels are significantly reduced in both genotypes upon cold exposure. Moreover, blood glucose levels are further reduced in cold-exposed AdRiKO mice (approximately -0.79mmol/L) compared to cold-exposed control mice. We believe that this minor difference in blood glucose levels between AdRiKO and control mice would not have altered the response to our exogenous 2-Deoxyglucose-tracer experiment.

4. The authors should provide information on the body weights of the mice analyzed as well as the BAT fat pad weights. This is important because many assays incl. O₂ consumption and glucose uptake use normalization to tissue weight.

Reply: We have now measured body weight and BAT weight of AdRiKO and control mice. These data are presented in Figures EV 1B and 2A of the revised version of the manuscript. Importantly, we did not observe any differences in either body weight or BAT fat pad weight between control and AdRiKO mice.

5. The authors should discuss the fact that the mice were fasted during cold exposure, meaning that mTORC2 function may only be required for thermogenesis under conditions of limiting substrates. Alternatively, the authors should provide experimental data to clarify this point (cold exposure with access to food).

Reply: To answer this question we have now performed cold exposure in control and AdRiKO mice with *ad libitum* access to food. Interestingly, while AdRiKO mice were still unable to maintain their body temperature with *ad libitum* access to food, the phenotype was less severe compared to the phenotype observed without access to food. These new data are presented in Figure EV 1I of the revised manuscript. Thus, as Referee #3 correctly predicted, these findings show that mTORC2 function in BAT is particularly important under conditions of limiting substrate availability. In light of this, we now discuss this important point in the results section of the revised manuscript.

6. The authors should indicate the number of replicates or mice in each figure legend. It is not possible to assess the statistical robustness of the results without the n numbers.

Reply: We apologize for this mistake and we have now added the number of replicates or mice in each figure legend.

Minor points

7. Literature references on the meaning of the different phosphorylation sites of Akt and their relation to mTORC2 should be provided at latest in the first results section (including a reference proving that the S473 site is a reliable marker for mTORC2 activity).

Reply: In response to this comment, we have now mentioned in the results section of the revised manuscript that Akt S473 phosphorylation is a reliable marker for mTORC2 activity, and provided the appropriate references.

8. The authors should specify the time point of analysis (duration of treatment/challenge) in the figure legends (Fig 1B-D, Fig. 2A-C, Fig. 3B,C, 4A,B, Fig. 5C-E, Fig. 6B-F).

Reply: We have now added the time points of analysis in the figure legends of above-mentioned figures.

9. Page 5 last sentence: the authors should correct "NE is released from the brain". NE is released in the periphery by sympathetic nerves (even if the activation of the nerves is initiated in the brain).

Reply: We have now rephrased the sentence to "NE is released from sympathetic nerves".

10. How was the "maximal respiration" defined and determined? This info is required in the Methods section.

Reply: For determination of maximal respiration the highest measured VO_2 value during the experiment for each mouse was used. We have now explained this in methods section of the revised manuscript.

11. What is the difference in the two panels of Fig. 6D? Significance tests should be performed and indicated for the data in the left panel. The same point applies to Fig. 6G.

Reply: We apologize if the representation of the data presented in Figures 6D and 6G was unclear. The left panel represents body temperature after each hour of cold exposure while the right panel represents body temperature as a bar graph for the 3h cold exposure time point. We have now explained this in the figure legends of the revised version of the manuscript. Moreover, we have also included significance signs in the left panel of Figures 6D and 6G.

References

Almahariq, M., Tsalkova, T., Mei, F.C., Chen, H., Zhou, J., Sastry, S.K., Schwede, F., and Cheng, X. (2013). A novel EPAC-specific inhibitor suppresses pancreatic cancer cell migration and invasion. *Mol Pharmacol* 83, 122-128.

Cybulski, N., Polak, P., Auwerx, J., Ruegg, M.A., and Hall, M.N. (2009). mTOR complex 2 in adipose tissue negatively controls whole-body growth. *Proceedings of the National Academy of Sciences of the United States of America* 106, 9902-9907.

Garton, A.J., Campbell, D.G., Cohen, P., and Yeaman, S.J. (1988). Primary structure of the site on bovine hormone-sensitive lipase phosphorylated by cyclic AMP-dependent protein kinase. *FEBS Lett* 229, 68-72.

Gasparetti, A.L., de Souza, C.T., Pereira-da-Silva, M., Oliveira, R.L., Saad, M.J., Carneiro, E.M., and Velloso, L.A. (2003). Cold exposure induces tissue-specific modulation of the insulin-signalling pathway in *Rattus norvegicus*. *J Physiol* 552, 149-162.

Gerhart-Hines, Z., Feng, D., Emmett, M.J., Everett, L.J., Loro, E., Briggs, E.R., Bugge, A., Hou, C., Ferrara, C., Seale, P., et al. (2013). The nuclear receptor Rev-erb α controls circadian thermogenic plasticity. *Nature* 503, 410-413.

Holz, M.K., and Blenis, J. (2005). Identification of S6 kinase 1 as a novel mammalian target of rapamycin (mTOR)-phosphorylating kinase. *The Journal of biological chemistry* 280, 26089-26093.

Hsieh, A.C., Liu, Y., Edlind, M.P., Ingolia, N.T., Janes, M.R., Sher, A., Shi, E.Y., Stumpf, C.R., Christensen, C., Bonham, M.J., et al. (2012). The translational landscape of mTOR signalling steers cancer initiation and metastasis. *Nature* 485, 55-61.

Hung, C.M., Calejman, C.M., Sanchez-Gurmaches, J., Li, H., Clish, C.B., Hettmer, S., Wagers, A.J., and Guertin, D.A. (2014). Rictor/mTORC2 loss in the *Myf5* lineage reprograms brown fat metabolism and protects mice against obesity and metabolic disease. *Cell reports* 8, 256-271.

Misra, U.K., and Pizzo, S.V. (2012). Upregulation of mTORC2 activation by the selective agonist of EPAC, 8-CPT-2Me-cAMP, in prostate cancer cells: assembly of a multiprotein signaling complex. *Journal of cellular biochemistry* 113, 1488-1500.

Morita, M., Gravel, S.P., Chenard, V., Sikstrom, K., Zheng, L., Alain, T., Gandin, V., Avizonis, D., Arguello, M., Zakaria, C., et al. (2013). mTORC1 controls mitochondrial activity and biogenesis through 4E-BP-dependent translational regulation. *Cell metabolism* 18, 698-711.

Mullins, G.R., Wang, L., Raje, V., Sherwood, S.G., Grande, R.C., Boroda, S., Eaton, J.M., Blancquaert, S., Roger, P.P., Leitinger, N., et al. (2014). Catecholamine-induced lipolysis causes mTOR complex dissociation and inhibits glucose uptake in adipocytes. *Proceedings of the National Academy of Sciences of the United States of America* 111, 17450-17455.

Nedergaard, J., and Cannon, B. (2013). UCP1 mRNA does not produce heat. *Biochimica et biophysica acta* 1831, 943-949.

Olsen, J.M., Sato, M., Dallner, O.S., Sandstrom, A.L., Pisani, D.F., Chambard, J.C., Amri, E.Z., Hutchinson, D.S., and Bengtsson, T. (2014). Glucose uptake in brown fat cells is dependent on mTOR complex 2-promoted GLUT1 translocation. *The Journal of cell biology* 207, 365-374.

Ribar, T.J., Rodriguiz, R.M., Khiroug, L., Wetsel, W.C., Augustine, G.J., and Means, A.R. (2000). Cerebellar defects in Ca²⁺/calmodulin kinase IV-deficient mice. *The Journal of neuroscience : the official journal of the Society for Neuroscience* 20, RC107.

Song, Y., Zheng, D., Zhao, M., Qin, Y., Wang, T., Xing, W., Gao, L., and Zhao, J. (2015). Thyroid-Stimulating Hormone Increases HNF-4alpha Phosphorylation via cAMP/PKA Pathway in the Liver. *Sci Rep* 5, 13409.

Tan, Y., Rouse, J., Zhang, A., Cariati, S., Cohen, P., and Comb, M.J. (1996). FGF and stress regulate CREB and ATF-1 via a pathway involving p38 MAP kinase and MAPKAP kinase-2. *The EMBO journal* 15, 4629-4642.

Thoreen, C.C., Chantranupong, L., Keys, H.R., Wang, T., Gray, N.S., and Sabatini, D.M. (2012). A unifying model for mTORC1-mediated regulation of mRNA translation. *Nature* 485, 109-113.

Vergnes, L., Chin, R., Young, S.G., and Reue, K. (2011). Heart-type fatty acid-binding protein is essential for efficient brown adipose tissue fatty acid oxidation and cold tolerance. *The Journal of biological chemistry* 286, 380-390.

Xing, J., Kornhauser, J.M., Xia, Z., Thiele, E.A., and Greenberg, M.E. (1998). Nerve growth factor activates extracellular signal-regulated kinase and p38 mitogen-activated protein kinase pathways to stimulate CREB serine 133 phosphorylation. *Molecular and cellular biology* 18, 1946-1955.

Zhang, H.H., Lipovsky, A.I., Dibble, C.C., Sahin, M., and Manning, B.D. (2006). S6K1 regulates GSK3 under conditions of mTOR-dependent feedback inhibition of Akt. *Molecular cell* 24, 185-197.

Appended Figure 1

A

B

C

D

E

Appended Figure 2

A

B

C

Appended Figure Legends

Appended Figure 1. (A) Immunoblot analysis of BAT cells stimulated with NE for 5 minutes in the presence of DMSO or 2 μ M H89 for the indicated proteins. (B) Immunoblot analysis of BAT cells stimulated with 8-Br-cAMP for the indicated proteins. (C) Immunoblot analysis of BAT cells stimulated with Isoproterenol (Iso) for the indicated proteins. (D) Accumulation of 2DG6P in BAT cells stimulated with NE for 2h in the presence of Rapamycin (Rapa) or Torin. (E) Appearance of AdRiKO and control mice.

Appended Figure 2. (A) Immunoblot analysis of isolated plasma membranes from BAT of AdRiKO and control mice treated with vehicle or insulin for the indicated proteins. Bar graph represents quantification of band intensity of GLUT4 (n \geq 3/group). (B) Appearance of differentiated primary brown adipocytes isolated from AdRiKO or control mice at day 5 after differentiation initiation. (C) Immunoblot analysis of BAT from AdRiKO and control mice housed at 22 $^{\circ}$ C or 4 $^{\circ}$ C for 2h for the indicated proteins (n=6).

Thank you for the submission of your revised manuscript to EMBO Molecular Medicine. We have now received the enclosed reports from the three referees who were asked to re-assess it. As you will see the reviewers are now globally supportive and I am pleased to inform you that we will be able to accept your manuscript pending the following final amendment:

Please address the minor text change commented by referees 1 and 2; we encourage you to reformulate the text as suggested. In addition, we would really appreciate if you could provide correct units as requested twice now by referee 1. Please provide a letter INCLUDING the reviewer's reports and your detailed responses to their comments (as a Word file).

Please submit your revised manuscript within two weeks. I look forward to seeing a revised form of your manuscript as soon as possible.

***** Reviewer's comments *****

Referee #1 (Comments on Novelty/Model System):

It is still my considered opinion that the cell line does not provide relevant information that is applicable to the in-vivo situation. As also pointed out by Referee 2, the signal transduction system may well be altered and therefore be irrelevant or misleading.

Referee #1 (Remarks):

The authors have responded extensively to all my comments, if not always, in my opinion, fully satisfactorily. However, I am prepared to accept their comments, except for the oxygen consumption - see bottom of p.6 and p.7 in the responses.

The authors argue that two other manuscripts used the same technique. Let me say only that this still does not make it correct. Nor does the fact that the authors saw an increase in OCR in tissue from 4 {degree sign}C animals mean that this is other than a qualitative indication of the presence of more mitochondria in those samples. I maintain that it does not accurately reflect cellular oxygen consumption capacity and I request that the authors formulate the text to reflect the qualitative rather than quantitative nature of the results.

I asked the authors to give the oxygen consumption data in Fig. 4I as ml per kg lean body mass or, in the absence of these data, then per mouse. The authors argue that because the body weights etc. were similar, they do not need to do this. I beg to differ. The units should be correct because other people follow printed data and do the same (note my comment above!), and other people will not always have animals of the same body composition. Therefore, the units should be correct, even if it does not make any apparent difference in the conclusions.

Referee #2 (Comments on Novelty/Model System):

The authors provide an interesting mechanism how glucose transport is maintained during catabolic conditions of cold exposure. The text is clear and very well written and most of the experiments support the conclusion claimed by the authors. Now the authors prepared a very careful and thorough revision.

Referee #2 (Remarks):

The authors have done a very careful and thorough revision and my points have been addressed. In principle, I recommend the acceptance of the paper but I still have one minor concern.

The authors state on page 10 of the revised manuscript that: 'Despite the drop in circulating insulin, glucose uptake into BAT was strongly increased upon cold exposure (Fig 5A). Thus, insulin plays only a minor role in cold induced glucose uptake (Shibata et al., 1989) and changes in circulating insulin are unlikely to explain the defect in cold-induced glucose uptake in BAT in AdRiKO mice.'

I strongly disagree that a drop in insulin levels indicates that it is unimportant for glucose disposal into cold-activated BAT. In my opinion this observation just indicates increased insulin sensitivity. In addition, there are several examples in the literature providing direct evidence that insulin signalling is involved in glucose disposal into BAT. For instance, it has been shown that alloxan treated rats with type 1 diabetes were unable to maintain body temperature in the cold, probably because of a failure to generate an adequate amount of heat by adaptive thermogenesis in brown adipose tissue. Thus, I would strongly encourage the authors to tone down their statement.

2nd Revision - authors' response

10 December 2015

Point-by-point response

We thank the referees for the positive evaluation of our manuscript and we are pleased to hear that we could satisfyingly address most of the referee's concerns. Please see below for the specific point-by-point reply to all remaining concerns.

Referee #1 (Comments on Novelty/Model System):

It is still my considered opinion that the cell line does not provide relevant information that is applicable to the in-vivo situation. As also pointed out by Referee 2, the signal transduction system may well be altered and therefore be irrelevant or misleading.

Referee #1 (Remarks):

The authors have responded extensively to all my comments, if not always, in my opinion, fully satisfactorily. However, I am prepared to accept their comments, except for the oxygen consumption - see bottom of p.6 and p.7 in the responses.

The authors argue that two other manuscripts used the same technique. Let me say only that this still does not make it correct. Nor does the fact that the authors saw an increase in OCR in tissue from 4 {degree sign}C animals mean that this is other than a qualitative indication of the presence of more mitochondria in those samples. I maintain that it does not accurately reflect cellular oxygen consumption capacity and I request that the authors formulate the text to reflect the qualitative rather than quantitative nature of the results.

Reply: We agree with referee #1 that it is not possible to explain the reason for the increase in respiration with the OCR measurement only. However, we would like to note that we have measured both mitochondrial DNA and protein content in AdRiKO and control mice housed at 22°C and 4°C (Fig. 4D and 4E). We could not observe an increase in mitochondrial DNA and protein content upon acute cold exposure in BAT of either control or AdRiKO mice. This suggests that the increase in OCR observed at 4°C most likely does not stem from the presence of more mitochondria in the tissue. However, following the suggestion of referee #1 we have now added a sentence in the results section that should reflect the qualitative nature of the results more clearly.

I asked the authors to give the oxygen consumption data in Fig. 4I as ml per kg lean body mass or, in the absence of these data, then per mouse. The authors argue that because the body weights etc. were similar, they do not need to do this. I beg to differ. The units should be correct because other people follow printed data and do the same (note my

comment above!), and other people will not always have animals of the same body composition. Therefore, the units should be correct, even if it does not make any apparent difference in the conclusions.

Reply: We have now corrected the units in Fig. 4H and 4I to mL/min/mouse since no data for body composition are available for this experimental cohort.

Referee #2 (Comments on Novelty/Model System):

The authors provide an interesting mechanism how glucose transport is maintained during catabolic conditions of cold exposure. The text is clear and very well written and most of the experiments support the conclusion claimed by the authors. Now the authors prepared a very careful and thorough revision.

Referee #2 (Remarks):

The authors have done a very careful and thorough revision and my points have been addressed. In principle, I recommend the acceptance of the paper but I still have one minor concern.

The authors state on page 10 of the revised manuscript that: 'Despite the drop in circulating insulin, glucose uptake into BAT was strongly increased upon cold exposure (Fig 5A). Thus, insulin plays only a minor role in cold induced glucose uptake (Shibata et al., 1989) and changes in circulating insulin are unlikely to explain the defect in cold-induced glucose uptake in BAT in AdRiKO mice.'

I strongly disagree that a drop in insulin levels indicates that it is unimportant for glucose disposal into cold-activated BAT. In my opinion this observation just indicates increased insulin sensitivity. In addition, there are several examples in the literature providing direct evidence that insulin signalling is involved in glucose disposal into BAT. For instance, it has been shown that alloxan treated rats with type 1 diabetes were unable to maintain body temperature in the cold, probably because of a failure to generate an adequate amount of heat by adaptive thermogenesis in brown adipose tissue. Thus, I would strongly encourage the authors to tone down their statement.

Reply: We thank referee #2 for this insightful remark and following the suggestion of referee #2, we have now removed our statement regarding the importance of insulin for cold-induced glucose uptake in the results section of the manuscript.